#### Evaluation of biases and uncertainties in ROMEX radio occultation observations

Richard Anthes, Jeremiah Sjoberg, Jon Starr, and Zhen Zeng

University Corporation for Atmospheric Research Constellation Observing System for Meteorology, Ionosphere and Climate (COSMIC) Program

Correspondence: Richard Anthes (anthes@ucar.edu)

#### **Abstract**

The Radio Occultation Modeling Experiment (ROMEX) is an international collaboration to test the impact of varying numbers of radio occultation (RO) profiles in operational numerical weather prediction (NWP) models. An average of 35,000 RO profiles per day for September-November 2022 from 13 different missions are being used in experiments at major NWP centers. This paper evaluates properties of ROMEX data, with emphasis on the three largest datasets: COSMIC-2 (Constellation Observing System for Meteorology, Ionosphere and Climate-2 or C2), Spire, and Yunyao.

The penetration depths (percent of profiles reaching different levels above the surface) of most of the ROMEX datasets are similar, with more than 80% of all occultations reaching 2 km or lower and more than 50% reaching 1 km or lower.

The relative uncertainties of the C2, Spire, and Yunyao bending angles and refractivities are estimated using the three-cornered hat method. They are similar on the average in the region of overlap (45°S-45°N). Larger uncertainties occur in the tropics compared to higher latitudes below 20 km. Relatively small variations in longitude exist.

We investigate biases in the observations by comparing them to each other and to models. C2 bending angles appear to be biased by about 0.15% compared to Spire and other ROMEX data between 10 and 30 km altitude. These biases, most of which are representativeness or sampling differences, are caused by the different orbits of C2 and other ROMEX missions around the non-spherical Earth and the associated varying radii of curvature.

### 1 Introduction

Deleted: apparent

Radio occultation (RO) observations have been shown to be among the top five observation types contributing to the accuracy of numerical weather prediction (NWP) forecasts with approximately 10,000 RO vertical profiles (atmospheric soundings) per day globally distributed (Anthes et al. 2024, hereafter A2024). Model simulation studies have shown a continued increase in positive impact of RO observations as the number of profiles increases to more than 100,000 profiles per day (Harnisch et al. 2013; Privé et al. 2022). In the near future, over 100,000 occultations per day may be available through commercial sources, offering the potential for further increases in forecast accuracy.

Until recently, when large numbers of commercial RO data became available, it has been impossible to test the impact of increasing numbers of RO profiles per day using real data beyond about 10,000 profiles/day. With the emergence of several private companies in the U.S. and China in the past few years, it became possible to acquire approximately 35,000 RO profiles per day for a three month period (September-October 2022) for testing in NWP models in the Radio Occultation Modeling EXperiment (ROMEX). ROMEX is being carried out under the auspices of the WMO International Radio Occultation Working Group (IROWG, <a href="https://irowg.org/">https://irowg.org/</a>). A2024 introduces ROMEX and reviews previous studies of the impact of RO observations on NWP forecast models. Shao et al. (2025) provide a summary of the IROWG tenth meeting (IROWG10) in September 2024 in which many initial ROMEX results were presented.

The ROMEX data became available at the European Organisation for the Exploitation of Meteorological Satellites (EUMETSAT) Radio Occultation Meteorology (ROM) Satellite Application Facility (SAF) in February 2024, and since then many international NWP centers have been testing the impact of these observations. This paper describes the characteristics of the ROMEX data, including depth of penetration into the lower troposphere, the standard deviation of random errors (uncertainties), and biases. We do not present any NWP results. However, because initial experiments in some of the NWP models using this unprecedented number of RO data showed a small degradation of model biases, we examine the ROMEX observation biases in detail.

 Table 1 in A2024 shows the average number of RO profiles per day from the 13 different missions. Of the total average number of 34,520 profiles per day, 78.4% are contributed by three missions: COSMIC-2 (4,900), Spire (16,750), and Yunyao (5,400). Therefore, in this paper we examine these three missions especially closely, because they are the ones likely to have the most impact on models. Furthermore, they are quite independent missions, representing one government mission (COSMIC-2) and two commercial missions from different countries, Spire (Europe and the US) and Yunyao (China). The satellites, orbits, instruments, and initial processing of these raw data are all different and independent. For brevity, we call this combined dataset CSY. Of the three datasets, C2 and Spire are relatively well known and have been widely studied (e.g. Schreiner et al. 2020; Bowler 2020), while Yunyao is a relatively new mission and has been under evaluation only recently. Cheng Yan (Yunyao Aerospace Technology Corp.) presented an introduction to the Yunyao mission and data at the 1st ROMEX workshop held at EUMETSAT in Darmstadt, Germany 17-19 April 2024 (Cheng 2024).

Deleted: 5

Preliminary results presented at the workshop indicated that the quality of Yunyao data after quality control (QC) was similar to that of other missions with some exceptions that were related to their suboptimal data processing and have since been corrected (Xu et al. 2025; Cheng 2025). A second Chinese commercial RO mission, Tianmu, was just getting started in 2022 and provided approximately 270 profiles per day for ROMEX. Almost a year later, at the 2<sup>nd</sup> ROMEX workshop at EUMETSAT 25-27 February 2025, both Yunyao and Tianmu presented results from greatly enhanced constellations, which were providing at that time 30,000 profiles per day from Tianmu (Qi Tang, 2025) and 33,000 profiles per day from Yunyao (Cheng, 2025). All presentations from the 1st and 2nd ROMEX workshops are available at <a href="irwg.org/romex-events-meetings/">irwg.org/romex-events-meetings/</a>.

# 1.1 Processing and analysis of ROMEX data

This section summarizes the methodology used to process the ROMEX data into bending angles, refractivities, and ultimately other products such as temperature and water vapor (not discussed here). The original (raw) data were downloaded from the satellites and processed independently into excess phase data by each data provider. A discussion of the fundamental RO observable excess phase and how it is used to derive the bending angle and refractivity is presented in The Radio Occultation Processing Package (ROPP) Pre-processor Module User Guide (<a href="https://rom-saf.eumetsat.int/romsaf">https://rom-saf.eumetsat.int/romsaf</a> ropp ug pp.pdf ).

Each provider used its own processing algorithms and QC. These are often proprietary for the commercial data and are not available. Because of the varying QC applied by each provider, it is important to compare the different datasets after applying additional QC that is uniform for all missions.

The excess phase data that passed the providers' QC were sent to EUMETSAT in January 2024, which then relayed them to two other processing centers, UCAR (University Corporation for Atmospheric Research) and NOAA STAR (Center for Satellite Applications and Research). EUMETSAT, UCAR, and STAR processed the excess phase data into bending angles, refractivities, and other products, as described generally by Kuo et al. (2004) and Steiner et al. (2020), using their own processing algorithms and QC. Because of NOAA policy, STAR does not process or distribute the Chinese data (Yunyao, Fengyun-3, and Tianmu).

Most of the NWP modeling centers have used the EUMETSAT-processed ROMEX data, which became available at the EUMETSAT ROM SAF in March 2024. Further information is available at <a href="https://irowg.org/ro-modeling-experiment-romex/">https://irowg.org/ro-modeling-experiment-romex/</a>. These data were all processed from the excess phases to bending angles and refractivities by EUMETSAT, except for C2, which were processed by UCAR. Since the data were provided to EUMETSAT in early 2024, more has been learned about their quality and processing and some of the ROMEX RO data have now been reprocessed and improved in quality. For example, Yunyao has improved some of the details of its processing, which was at an early stage in 2024. Recently (late 2024) a source of small biases in all ROMEX data was found by Aparicio (2024). He showed that the sideways

Deleted: 4

sliding of the RO occultation plane and tangent point can cause biases due to the variation of Earth's radius of curvature (radius of a sphere that best fits the Earth's surface curvature at a given location and orientation of the RO occultation plane and is used in the RO bending angle retrievals) and its subsequent effect on the height of the observation. Other small changes have likely been made by other providers to improve their RO data and products. However, in this paper we evaluate the bending angles (BA) and refractivities (N) in the level-2 BUFR products (BfrPrf) processed by UCAR from the ROMEX excess phase data that were originally provided to EUMETSAT. Details of the UCAR processing are described by Sokolovskiy (2021). Performing structural uncertainty analyses similar to Steiner et al. (2020), in limited comparisons we find that the UCAR-processed data and the EUMETSAT-processed data are similar in most respects; examples are shown in the Supplement (S9). A detailed comparison of the two datasets is being carried out by UCAR and EUMETSAT.

We estimate the lower tropospheric penetration depths (lowest level reached) of the RO profiles, the standard deviation of random errors (uncertainties), and biases. The penetration depths depend on the cutoff criteria used in the processing, and so their comparison among different missions should be done with the same processing center.

Radio occultation observations (X) can be written as Truth (T) plus a bias (b) and random error  $(\varepsilon)$ :

$$X = T + b + \epsilon \tag{1}$$

The variance of the random errors is given by

$$Var (\epsilon) = Var(X-T-b) = \langle \epsilon^2 \rangle$$
 (2)

where < > is the sample mean. The standard deviation (STD) of the error is the square root of the variance.

The bias of a sample of observations is <X-T>. Truth is never known but, historically, RO observations have been considered to be largely unbiased above the lower troposphere because they are based on measurements of doppler shifts of the refracted signals using precise atomic clocks, which enables traceability to SI-traceable measurements of time (Leroy et al. 2006). RO observations are therefore assimilated in NWP models without bias corrections (Healy 2008; Cucurull et al. 2014) and have been shown in many studies to act as "anchor" observations in the model forecasts (e.g., Aparicio and Laroche 2015), improving the impact of radiance measurements, which must be bias corrected. However, several early forecast experiments reported at the April 2024 ROMEX workshop showed small negative impacts on the biases of model forecasts when ROMEX data were assimilated, even though most forecast skill metrics showed positive impacts. Estimates of biases in ROMEX datasets with respect to other ROMEX data sets indicated possible biases of order +/-0.2%. Such small biases are not

#### Deleted: ¶

Deleted: We estimate the lower tropospheric penetration depths of the RO profiles, the standard deviation of random errors (uncertainties), and the biases. The penetration depth is defined as the percentage of profiles in a sample of RO observations reaching different levels above the ground. The penetration depth (lowest level reached) depends on the cutoff criteria used in the processing, and so comparisons of the penetration rates of different missions should be done with data from the same processing center....

Deleted: ROMEX

easily visible in commonly used verification charts of (O-B)/B (normalized observations minus model background or a reference dataset), in which the relative biases and standard deviations of differences are often plotted together on a scale of -20% to +20% (e.g. Schreiner et al. 2020; Ho et al. 2023). The impact of ROMEX data on several model biases led to studies on possible sources of the model biases, including previously undetected small biases in the RO observations, model biases, biases in the forward model estimates of bending angle from the model data in the data assimilation process, suboptimal interactions with the bias correction of radiances, and small systematic errors in matching the heights of the model variables to the heights of the RO observations (1st and 2nd ROMEX workshops irowg.org/romex-events-meetings/, Shao et al. 2025).

RO uncertainties and biases are smallest in the upper troposphere and lower to middle stratosphere between approximately 8 and 35 km (Anthes et al. 2022) and the differences between RO missions and processing methods are also smallest in this layer, which is sometimes colloquially called the RO core region, golden zone, or sweet spot. Because of the small uncertainties and biases in this layer, RO observations are weighted most heavily in data assimilation and have the most impact on model analyses and forecasts in this layer (Ruston and Healy 2020). Therefore, in this study we primarily focus our attention on the 10-30 km layer.

Uncertainties and biases are estimated by comparing the ROMEX observations to other datasets. In this paper we use analyses or short-range forecasts from ECMWF (European Centre for Medium-range Forecasts) operational model, ERA5 (fifth generation ECMWF reanalysis; Hersbach et al. 2020), and JRA-3Q (Japanese Reanalysis for Three Quarters of a Century; Kasaka et al. 2024), and other RO data. Bending angles from the model were calculated using a 1D-forward model (Syndergaard et. al 2006; Gilpin et al. 2019). Biases and uncertainties in the model BA do not necessarily imply biases and uncertainties of similar magnitudes in the model temperature or water vapor. The BA are a function of the vertical gradient of these model variables, and may also arise from systematic errors in the forward model, such as errors in the coefficients of the refractivity equation.

In comparing different datasets, it is important to minimize sampling differences by collocating the data. When RO data are compared with other RO or radiosonde data, collocation is usually done by comparing samples of pairs of the two datasets close to each other in space and time, e.g. 300 km and 3 hours. The closer the collocation, the more the sampling differences are reduced (Nielsen et al. 2022), but at the expense of fewer pairs in the sample and greater noise in the statistics. For our analysis of collocated datasets, the sample sizes far exceed the sample size of order 1000 suggested by Sjoberg et al. (2021) where statistical noise in the three-cornered hat (3CH) method may be considered negligible. A reduction of the sampling difference between nearby but not perfectly collocated profiles may be achieved by double differencing using model data (Tradowsky et al. 2017; Gilpin et al. 2018). When RO observations are compared with model data, the model data may be interpolated to the actual time and location (tangent point) of each RO observation at each level,

Deleted: Perceived b

Deleted: necess

Deleted: may

**Deleted:** biases in the model data (e.g. temperature and water vapor) or ...

accounting for the tangent point drift, which may be 100 km or more. Use of a global model as the reference dataset enables many more collocations because model data are available at all times and locations globally. However, model data have different representations of the atmosphere (footprints), require forward models, and have their own biases. We also consider the global geographic variation of biases and uncertainties by binning the RO and model data into 5° latitude-longitude bins and averaging over the three-month period of ROMEX.

#### 1.2 Estimation of uncertainties

The uncertainties of the ROMEX observations are estimated by the 3CH method, which was developed many years ago to estimate the uncertainties in atomic clocks (Sioberg et al. 2021). In the 3CH equations, the error-free truth (T) does not appear. Sioberg et al. (2021) discuss the concept of truth in the context of the 3CH method, which is nontrivial as pointed out by O'Carroll et al. (2008). Most other studies estimate the error variance of a dataset X by approximating truth by an independent dataset Y (often a model background B) and the uncertainties are computed as the standard deviation of the differences between X and Y. The 3CH method uses three datasets (X, Y, and Z) and is slightly more accurate and has the advantage of providing estimates of the error variances of the other two datasets simultaneously (Anthes and Rieckh, 2018; Rieckh et al. 2021). It is equivalent to the Desroziers' method (Desroziers et al. 2005) under certain conditions (Semane et al. 2022; Todling et al. 2022), which is used by many modeling centers. Both methods of estimating the uncertainties assume independent datasets, i.e., negligible error covariances. Both methods also contain representativeness differences if the footprints (spatial and temporal scales represented by different observations) of the datasets differ (Sjoberg et al. 2021).

#### 1.3 Estimation of biases

Biases are more difficult to estimate than uncertainties because the truth is unknown. In addition, truth depends on the footprints of the observations. For example, truth for radiosondes, which are essentially point measurements, is different from truth for RO, which represents an average over a pencil-shaped volume of atmosphere approximately 250 km along the ray path and 1 km in diameter (Anthes et al. 2000). The biases of RO BA and N are estimated by comparing them to other datasets such as model analyses or reanalyses, radiosondes, or other RO observations, which are different proxies for truth. These bias estimates are always approximate, because the comparison datasets that are used as references have their own biases and there can be representativeness differences between the two datasets; we do not assume either dataset is truth. Thus, theoretical estimates of observation biases (e.g., Melbourne et al. 1994; Kursinski et al. 1997) together with comparisons to multiple independent and trusted datasets are useful to establish a likely range of observation biases.

As noted above, the biases of RO data in the upper troposphere and stratosphere are generally assumed to be zero and are assimilated without bias corrections in NWP models. Early studies estimated that the biases are very small. For example, John Eyre

Deleted: cigar

in a 2008 workshop (Eyre 2008) estimated that systematic errors in temperature were less than 0.2 K, noting that this value was to be demonstrated. For a temperature of 270 K, 0.2 K is 0.07%. It has been difficult to demonstrate such a small bias in subsequent studies, and even a bias of 0.1% is important in climate studies (Steiner et al. 2020; Ho et al. 2024). We take a close look at biases in the ROMEX data in later sections of this paper.

# 2 Overall properties of ROMEX observations

In some of our results, we compare bending angle bias and uncertainty profiles of the ROMEX missions as a function of impact height, which is related to the geometric height by the refractivity and local radius of curvature of the Earth (Sokolovskiy 2010). The influence of the occultation plane's azimuth angle on these comparisons, discussed in Section 5, results in representativeness differences that are not differences in the quality of the retrievals. The magnitude of these differences (less than 0.15%) is much smaller than the 3CH uncertainty estimates, which are 1.5% or higher. However, they may have an impact on the comparison of bending angle biases, which are of the same order of magnitude between 10 and 30 km.

#### 2.1 Geographic and local time coverage

The profile counts of the 13 different missions (sources) of ROMEX data are provided in A2024. Figure 12 of A2024 shows the global coverage of all ROMEX data on one day, as well as the local time coverage on this day. The geographic coverage is quite uniform, but because many of the satellites are in similar polar orbits, the number of profiles is maximum between 09:00-12:00 and 21:00-00:00 local times, with other local times showing considerably fewer observations.

Fig. 1 shows the local time coverage of C2, Spire, and Yunyao, and the combined dataset CSY for 1 September 2022. The local time coverage is concentrated between 09:00-12:00 and 21:00-00:00 for Spire, and around noon and midnight for Yunyao. C2 is restricted to tropical and subtropical latitudes but covers all local times fairly uniformly. The combined local time coverage shows maximum coverage at about 10:00 and 22:00 and minimum coverage at about 06:00 and 18:00.

Fig. 2 shows how the non-uniform local time coverage for 1 September 2022 affects the distribution of observations in six-hour UTC time windows, which is the typical data assimilation cycling window in NWP models (e.g., NOAA's Global Forecast System or GFS). The colors represent the age of the observation received in each 6-h window. The youngest observations have more impact than the oldest observations (McNally 2019). The maximum cluster of young observations sweeps westward during the day, occurring over the Pacific and Atlantic Oceans around 00 and 12 UTC. Although the CSY data (and the ROMEX total) provide well-distributed global coverage over a 24-h period, the local time coverage is not uniform, with relative gaps occurring around 06:00 and 18:00. This uneven distribution will likely have some impact when high-impact weather events (such as tropical cyclones) are developing at times of relatively sparse

coverage (gaps in local time) but is not expected to have a large impact on the three-month statistics.

The distribution of ROMEX data for one day over a high-impact regional weather event (Hurricane Ian, 2022) is shown in Fig. 3. This figure indicates that the 35,000 ROMEX profiles per day have adequate coverage to resolve the large-scale structure of important weather phenomena such as tropical cyclones. Many studies have shown the RO observations can make a major improvement in TC genesis and track forecasts (Chen et al. 2022 and references therein).

Fig. 4 shows the total counts of CSY, Yunyao, Spire, and C2 in 5° latitude-longitude bins over the 3-month period of ROMEX. The C2 counts are smallest (fewer than 100) in the 40-45° NS (40-45° north and 40-45° south) bins, which means that on some days there may be only a few C2 observations in a bin at these latitudes and sampling issues may arise. The undulating minimum in counts of Spire near the Equator corresponds to the ionospheric Equatorial anomaly (Caldeira et al. 2020) and was first pointed out by Chris Barsoum (Aerospace Corporation, personal communication February 2025). This minimum indicates a higher rejection rate of Spire observations in the Equatorial anomaly. It does not appear in the C2 observation counts, probably related to the different orbits, signal to noise ratio, and other aspects of the two missions.

The total number of the C2, Spire, Yunyao, and CSY profiles for 0.1° latitude bands for the entire ROMEX period is shown in Fig. 5 from two different perspectives. The left panel shows total number vs. cos(latitude) while the right panel shows the total number density per 10,000 square km. The distributions of C2 (low-inclination orbits) complement the distributions of Spire and Yunyao, which are in high-inclination orbits.

Fig. 1: Local time coverage of Spire, Yunyao, COSMIC-2, and CSY (combined COSMIC-2, Spire and Yunyao) for 1 September 2022. The x-axes are local time in hours. The map background is included to help visualize the scale of the gaps. These are UCAR-processed data that have passed the CDAAC (COSMIC Data Analysis and Archive Center) QC. Figure prepared by Valentina Petroni, UCAR COSMIC Program.

\$75

Fig. 2: Six-hourly distributions of CSY for one day (1 September 2022): 00-06 UTC (top left), 06-12 UTC (top right), 12-18 UTC (lower left), and 18-24 UTC (lower right). Colors indicate age of observation at the end of each six-hour window (red 0-2h, orange 2-4h, green 4-6 h). The youngest observations (red) have the most impact in the 6-h data assimilation cycle. These are UCAR-processed data that have passed QC. Figure prepared by Valentina Petroni.

Fig. 3: All ROMEX data in one day (27 September 2022) superimposed on a GOES-16 geocolor image from 17:00 UTC. These are UCAR-processed data that have passed QC. Figure prepared by Valentina Petroni.

Fig. 5: Number of profiles over the 3-month ROMEX period (x-axis) in 0.1° latitude bins for C2 (red), Spire (blue), Yunyao (green), and combined CSY (black). The panel on the left is count vs. cos(latitude). Panel on right is count per 10,000 square km vs. latitude.

# 2.2 Numbers and stability of CSY observations over ROMEX time period

Fig. 6 shows the daily BA profile counts after CDAAC QC but before the final QC as described in Section 2.3, 3CH uncertainties, and biases with respect to ERA5 at 20 km for C2, Spire, Yunyao, and CSY over the ROMEX period. All three missions, but especially Spire and Yunyao, show large fluctuations in counts from day to day. However, the statistics (biases and uncertainties) are fairly constant and are similar for the three missions. Biases are slightly positive for C2 and slightly negative for Spire and Yunyao. Latitudinal sampling differences between C2 and the two polar-orbiting missions Spire and Yunyao are significant in these comparisons of biases and uncertainties.

Deleted: 3CH

Fig. 6: Number of occultations per day (dotted lines) and error statistics (uncertainties in solid and biases with respect to ERA5 in dot-dashed) of BA for C2 (red), Spire (blue), Yunyao (green), and CSY (orange) at 20 km. The CSY daily counts are not shown. The uncertainties and biases are normalized by the sample mean of ERA5.

# 2.3 Quality control and frequency distribution of CSY data

423 424

429

431

In addition to the QC applied by the providers on the original excess phase data and by UCAR in the processing of these data to bending angles and refractivity, we provide a final QC on the BA and N before evaluating the uncertainties and biases. We first check on super refraction (SR) based on collocated model data and remove any RO data for which the collocated model data indicate SR (vertical refractivity gradients exceeding -157 N units/km). This QC does not necessarily remove all RO observations with SR. We then remove outliers based on departures of the individual observations from the collocated ERA5 data, analogous to the (O-B)/B QC applied by operational NWP centers in their assimilation process. Our reasoning was that the highest and lowest BA were not necessarily the lowest quality, but rather the observations farthest from a trusted dataset were more likely to be of dubious quality. Our QC removes the highest and lowest 0.1 percentile of the (O-ERA5)/ERA5 data. This QC step is applied to all three CSY datasets, and results in approximately 0.4% of the observations removed. The resulting distributions of the BA values and (O-ERA5)/ERA5 at several different levels during the ROMEX period is shown in Fig. 7. The distributions of the BA observations are far from normal, reflecting the non-normal frequency of common atmospheric patterns at different levels, especially near the tropopause (20 km) where there are three distinct maxima. However, the frequency distributions of the (O-ERA5)/ERA5 data are nearly normal at all levels.

Fig. 7: Frequency distributions of CSY ROMEX data after QC at different levels (3, 5, 10, 20, 30, and 50 km impact height). The top panel at each level is the distribution of BA values in microradians and the lower panel at each level is the distribution of (O-ERA5)/ERA5 values.

# 2.4 Penetration depths

RO profiles penetrate to different levels above the surface, depending on the way the data are processed (how the lower cutoff is determined and quality control) and atmospheric conditions. The latter is especially important, as penetration depths are much lower (closer to the surface) with cool, dry atmospheres, and thus there are large variations with latitude. There is some evidence that higher signal-to-noise ratio (SNR) enables slightly lower penetrations (Schreiner et al. 2020).

Fig. 8 shows the penetration depths for all missions and latitudes. Most missions show more than 80% of all occultations reach 2 km or lower and more than 50% reach 1 km or lower. The penetration depths are noticeably less for Metop-B and -C (two shades of green, overlapping on this figure), Tianmu (light yellow), and Yunyao (purple). The penetration depths for these UCAR-processed Metop data are noticeably higher than those for the EUMETSAT-processed data, which is likely an artifact of the UCAR processing and is being investigated. The penetration rates for COSMIC-2 and Spire are very similar, in spite of the higher SNR for COSMIC-2. These results confirm that radio occultation is a useful method of obtaining global information on the planetary boundary layer (Ao et al. 2012).

Fig. 8: Fractional count of penetration depth for all ROMEX missions (all latitudes top left and 45°NS top right) and COSMIC-2, Spire, and Yunyao (all latitudes bottom left and 45°NS bottom right). Figure prepared by Hannah Veitel, UCAR COSMIC Program.

# 3 Overall bias and uncertainty statistics of ROMEX data

487

488 489

493

497

507

510

513

In this section we present an overview of the bias and uncertainty statistics of all the ROMEX data. Many additional figures showing statistics for the three largest ROMEX datasets are presented in the Supplement. Fig. 9 shows the biases and standard deviations of ROMEX differences from ECMWF analyses vs. mean sea level (MSL) altitude. The ECMWF data are interpolated to the time and place of the RO tangent point, accounting for tangent point drift. We note that the ECMWF analyses contain an impact of some, but not most, of the ROMEX data, because they assimilated the operational RO data of this time period (approximately 7,000-7,500 profiles per day). Despite quite different latitudinal sampling, the uncertainties and biases of the ROMEX data are similar between about 8 and 35 km MSL height, where RO observations have the most impact on NWP forecasts. The uncertainties vary most strongly above 40 km. with Sentinel-6, Metop-B, and Metop-C having the smallest uncertainties because of their more accurate clocks (Bonnedal et al. 2010, Padovan et al. 2024). Fengyun-3 shows higher uncertainties between 10 and 30 km than the other missions. Yunyao has a peak in uncertainties between 10 and 15 km, which is associated with their initial nonoptimal processing as discussed earlier.

Fig. 9: Biases and standard deviations of differences from ECMWF analysis for all ROMEX missions. All latitudes are included. Figure prepared by Hannah Veitel.

The biases of all ROMEX missions with respect to ECMWF analyses appear very close to zero on this scale of the x-axis (Fig. 9), but a closer look shows a small negative bias of approximately -0.1% in most ROMEX missions between 10 and 35 km as shown in Fig. 10a, COSMIC-2, however, shows a small positive bias of approximately 0.1-0.15%. When the large number of ROMEX data are assimilated in models, biases of this order

Deleted: (

Deleted: )

of magnitude could reveal issues in the NWP models that were not apparent when smaller numbers were assimilated. We examine these small biases in greater detail in Sections 5.2 and 6.

Fig. 10: (a) Mean differences of bending angles of all ROMEX missions from ECMWF analyses from 10 to 50 km MSL altitude, all latitudes included. (b) Close up of biases of all C2, Spire, and Yunyao (all latitudes included). (c) Biases of C2, Spire, and Yunyao, 45°NS only. (d) Biases of C2, Spire, and Yunyao, 30°NS only. Figure prepared by Hannah Veitel.

When all latitudes are considered together, the Spire and Yunyao biases are negative compared to C2 by about 0.2% between 15 and 35 km (Fig. 10b). However, this relatively large difference is primarily because all latitudes are being compared, and there are significant latitudinal sampling differences. When the data are restricted to the C2 latitudes of 45°NS only (Fig. 10c), the differences in the three missions are reduced to approximately 0.1%, as the biases of Spire and Yunyao are instead slightly positive at these latitudes. When the data are compared only between 30°NS (Fig. 10d), the C2 and Spire biases are nearly identical and only about 0.05% larger than Yunyao. These figures show the importance of comparing different RO missions using spatial sampling as similar as possible.

# 4 Detailed evaluation of COSMIC-2, Spire, and Yunyao

#### 4.1 Uncertainties

In this section we look at the 3CH uncertainties for the UCAR-processed C2, Spire, and Yunyao data, as well as the combined dataset (CSY). The other two datasets (corners) used in the 3CH method are short-range forecasts of ERA5 and JRA-3Q reanalyses,

and these model data are interpolated to the time and place of the RO observations, accounting for tangent point drift. We use short-range (6-18 h) forecasts verifying at the time of the analysis so that the models will not have assimilated the observations being analyzed and hence have minimum error correlations. However, ERA5 and JRA-3Q may have error correlations because they assimilate similar observations; such a correlation would lead to overestimates of the uncertainties of the RO observations being evaluated. We compare the statistics of the data at all latitudes as well as the data confined to 45°NS, where all the C2 data occur.

50

 Fig. 11 shows the normalized 3CH uncertainties of the CSY dataset (all latitudes). For comparison, the simple but effective RO observation error model used by ECMWF (Ruston and Healy 2022) is shown as a dashed line. Considering that it was developed many years ago, the agreement with the CSY data between 10 and 35 km is remarkable.

The 3CH uncertainties of the RO data are at a minimum between about 10 and 35 km impact height, averaging about 1.5% in this deep layer. They increase toward the surface, reaching a maximum of about 12% at an impact height of 3 km (geometric height about 1 km) and then decrease toward the surface to about 6%. Above 35 km the uncertainties increase rapidly, exceeding 40% above 55 km. Qualitatively the uncertainties from the 3CH method are similar to those of the standard deviations of the differences of the ROMEX and ECMWF data as shown in Fig. 9. The ERA5 uncertainties are the smallest of the datasets, especially above 30 km. The JRA-3Q uncertainties exceed the observations by a small amount in the lower troposphere, and then are slightly greater than the ERA5 data from 5 to 60 km.

Fig. 11: 3CH BA uncertainties of the CSY data. Also shown are the uncertainties of the two other corners of the 3CH method, ERA5 (blue) and JRA-3Q (green). The orange dashed curve, identified by ECMWF in the figure, is the ECMWF assumed RO observation error model (Ruston and Healy 2020). The data counts are given in gray.

Fig. 12 shows the 3CH uncertainties of C2, Spire, and Yunyao separately, for all latitudes (left) and 45°NS (right). The uncertainties of the 45°NS datasets are slightly larger below 10 km and slightly smaller above 30 km compared to the all-latitude uncertainties. Although Yunyao shows an anomalous increase between 10 and 15 km, the similarity of the uncertainties of these three independent RO datasets is remarkable and supports the use of a common relative RO error model for all missions as done by ECMWF. The anomalous feature in the Yunyao data between 10 and 15 km is related to Yunyao's transition from geometric to wave optics in their early processing and has been resolved in Yunyao's current processing (Xu et al. 2025).

Fig. 12: 3CH BA uncertainties for COSMIC-2 (red solid), Spire (red dash-dotted), and Yunyao (red dashed), and the two corners of the 3CH method ERA5 (blue) and JRA-3Q (green). There are three estimates for the error variances of ERA5 and JRA-3Q, one for each RO mission; the differences are small and <a href="mailto:barely.visible">barely.visible</a> in this plot. The dataset for all latitudes is shown in the left panel; the dataset for 45°NS is on the right. BFRPRF refers to the three RO missions. Above 30 km the Yunyao and C2 profiles are nearly indistinguishable in the left panel and in the right panel Spire and C2 are nearly indistinguishable, which illustrates the closeness of these three datasets at these levels.

Although the global 3CH relative uncertainties of the C2, Spire, and Yunyao BA observations are similar, there are variations in different geographic regions. Fig. 13 shows the 3CH uncertainty estimates for the combined dataset at 3 km, 5 km, 10 km, 20 km, 30 km, and 50 km computed in 1° latitude-longitude bins. Enlarged maps for the uncertainties of CSY and three datasets separately can be found in the Supplement. At 10 km and below the uncertainties are generally higher in the tropics and subtropics, but there is no simple geographic variation with latitude and longitude that describes the variations at all levels. An interesting regional feature is the maximum uncertainty over the Weddell Sea at 20 and 30 km, which may be related to the ionospheric Weddell Sea anomaly (Chang et al. 2015). The Weddell Sea anomaly is a recurrent feature of the austral summer midlatitude ionosphere where electron densities are observed to maximize during the local nighttime.

Deleted: not

Fig. 13: Global distribution of 3CH uncertainties (%) for CSY BA at 3 km (a), 5 km (b), 10 km (c), 20 km (d), 30 km (e), and 50 km (f). The color code denotes departures from global mean value at each level (denoted by white); blue represents below average uncertainties and red represents above average uncertainties. The color code is different for each level, and the range is an order of magnitude larger in the 50 km map (Fig. 9f). The zonal mean uncertainties are shown in plots to the left of each figure and the longitudinal means of the uncertainties are shown in plots at the bottom of each panel. Larger versions of the panels are presented in the Supplement (S3).

## 4.2 Biases

The small negative impact of the ROMEX data on the biases of several NWP models has caused intensive study of possible causes of these biases, including the possibility of biases in the ROMEX data (discussed in the two ROMEX workshops <a href="https://irowg.org/romex-1/">https://irowg.org/romex-1/</a> and <a href="https://irowg.org/romex-2/">https://irowg.org/romex-1/</a> and <a href="https://irowg.org/romex-2/">https://irowg.org/romex-1/</a> and <a href="https://irowg.org/romex-2/">https://irowg.org/romex-2/</a>). Indeed, it appears that most ROMEX data may have a small negative bias of approximately -0.15% between 10 and 30 km. Fig. 10a shows this bias with respect to ECMWF analyses, while Bowler (2024), Syndergaard and Lauritsen (2024), and Ho et al. (2024) found similar negative biases.

Deleted: small

Deleted: small

This section takes a close look at the biases of C2, Spire, and Yunyao, which appear to be between +/-0.15% between 10-30 km (Fig. 10b).

We estimate the biases of a sample of ROMEX data in two ways. The first way is to collocate each member of an RO dataset with a nearby member of a reference dataset (a model or another RO dataset) and compute the mean differences of the pairs, with advantages and limitations discussed in Section 1.1. In the second way we first locate each RO observation into a latitude-longitude grid (e.g. 5°x5°) at constant impact height levels over a specified time interval (we use two days, but the results are not sensitive to the time interval). The location of the RO observation is where the tangent point of the profile falls within the bin. We then compute the mean difference of each RO observation in the grid cell from the average value of the reference data (e.g., another RO dataset or a model) over the grid, denoted by <(RO-<Reference>)>, Finally, we average over all grid boxes and the time period of the sample (3 months) and normalize by the entire sample mean of the reference dataset, denoted by <<Reference>>. If the observations are located randomly within each grid box, sampling differences should cancel in the average, leaving only biases between the RO and the reference. There is no weighting of the data with latitude; it is merely a mean difference of a sample of RO observations compared to a reference dataset. This method has the advantage of using all RO observations in the sample rather than only those that have a nearby reference and also allows viewing geographical differences of the biases.

Fig. 14a shows vertical profiles of the bending angle biases of C2, Spire, and Yunyao compared to ERA5 short-range forecasts. The biases of Spire and Yunyao (blue and green profiles, respectively) are almost identical between 15 and 40 km, while the C2 biases (red profile) are slightly higher. Below about 4 km impact height, all three RO missions show a large negative bias in BA. These negative BA biases are also visible near the surface in all ROMEX missions (Fig. 9), as well as N (examples shown in Supplement). Large negative biases in BA below 4 km impact height in low latitudes are mainly related to wave propagation effects under strong horizontal and vertical N gradients induced by moisture (Sokolovskiy et al. 2010; Gorbunov et al. 2015). This bias propagates into N after the Abel inversion (Kursinski, 1997). When the vertical N gradient exceeds a critical value of -157 N-units per km, as it often does near the top of the atmospheric boundary layer, superrefraction occurs and the Abel inversion results in an additional negative N bias (Sokolovskiy 2003; Xie et al. 2006; Feng et al. 2020).

Deleted: two

Deleted: R

Fig. 14: (a) C2, Spire, and Yunyao bending angle biases vs. short-range (0-18 h) ERA5 forecasts computed from 5°x5° latitude-longitude bins averaged over all bins and days of ROMEX. (b) Biases of ROMEX CSY bending angles vs. short-range ERA5 forecasts computed from 5°x5° latitude-longitude bin averages over all bins and days of ROMEX. from 0 to 60 km impact height. (c) enlarged plot of 14(b) from 10 to 40 km. Note the change in range of the x-axis. Above 30 km, ERA5 biases are likely dominant (see text).

In Fig. 14a and 14b, the biases relative to ERA5 in the core region appear to be close to zero, as in Fig. 9 (reference ECMWF analysis). However, in the enlarged version (Fig. 114c), a negative bias of about -0.1% is evident between 10 and 25 km, similar to the negative bias of the entire ROMEX dataset (Fig. 10a). The positive biases beginning between about 35 km and the negative biases above 50 km, are likely due mainly to biases in ERA5, as indicated by the strong agreement of the three independent RO datasets in Fig. 14a. Biases in model BA and N may arise from biases in the model temperatures at these levels or systematic errors in the forward models used to compute the BA and N from the model data.

Fig. 15 shows Yunyao and C2 normalized BA biases relative to Spire between 10 and 40 km impact height. The close agreement of Yunyao and Spire between 15 and 40 km in Figs. 14a and 15, with average differences less than 0.1%, is remarkable given that the missions are independent commercial missions from two different countries. In contrast, C2 has a positive bias of about 0.1% relative to Spire. The bulge between 15 and 20 km in both the C2 and Yunyao profiles is likely related to the relatively large horizontal sampling differences in the 5°x5° latitude-longitude bins (Fig. 5) in a layer with large variations of atmospheric densities in the vicinity of the tropopause since this bulge is not evident when C2 and Spire are very closely collocated (Fig. 17).

Deleted: The b

Fig. 15: Yunyao and C2 BA biases relative to Spire between 10 and 40 km impact height. These are computed from 5°x5° latitude-longitude bin averages over all bins and days of ROMEX. Shown are Yunyao biases for all latitudes and for 45°NS only to more closely match C2.

The geographic distribution of the CSY BA biases relative to ERA5 at six levels is shown in Fig. 16. Larger versions of these figures and the corresponding CSY N biases are given in the Supplement. We note that the N biases above 40 km are affected by the statistical optimization, which can vary with different processing centers. These are computed from 1° latitude-longitude bins. Similar to the uncertainties (Fig. 13), the largest biases at 5 km, 10 km, and 20 km are located in the tropics. Regions of large biases at 5 km occur over the western Atlantic and South America, the western Pacific, Asia, and Indian Ocean, perhaps associated with regions of strong moist convection. Bands of negative or near-zero biases exist off the west coasts of South America and Africa at 5 km. At 30 km, biases are small. ERA5 biases may be of comparable or larger magnitude at all levels.

Deleted: apparent
Deleted:

Fig. 16: Global distribution of BA biases (%) relative to ERA5 short-range forecasts for CSY (combined C2, Spire, Yunyao) at 3 km (a), 5 km (b), 10 km (c), 20 km (d), 30 km (e), and 50 km (f). Larger versions of the panels with some comments are presented in the Supplement (S1).

## 5 Positive biases in COSMIC-2 between 10 and 30 km

In addition to the results shown here, several other, independent studies have indicated that C2 BA observations have a small positive bias between approximately 10 and 30 km compared to models and other RO data from polar-orbiting satellites. For example, a EUMETSAT report evaluating Sentinel-6 data showed a C2 positive bias of ~0.2% (EUMETSAT 2022, Fig. 33). Positive biases of C2 BA and N vs. ERA5 and other RO missions in the lower stratosphere have also been reported by Ho et al. (2024, 2025). The ROM SAF Matched Occultation page presents daily estimates of the biases of RO satellites compared to other RO satellites, with a collocation criteria of 300 km and 3 hours (<a href="https://rom-saf.eumetsat.int/monitoring/matched.php">https://rom-saf.eumetsat.int/monitoring/matched.php</a>). A comparison of C2 satellites with other satellites (e.g. Metop-B) shows a slight positive bias (about 0.1-0.2%) between about 10 and 30 km. Above 40 km and below 8 km the mean differences are larger, exceeding several percent; these will not be discussed further as

**Deleted:** This monitoring site shows mean and standard deviation of differences between BA and N from different satellites. ...

they are in layers that currently have small impact in NWP models. In this section we investigate the bias between 10 and 30 km in greater detail. For this discussion, we use Spire as an example of polar orbiting satellites – given its large data volume within ROMEX – to explain the observed positive C2 biases relative to other RO missions.

#### 5.1 C2 bending angle and refractivity biases relative to Spire

758

760

763

767

774

778

Fig. 17 illustrates the C2 biases in BA and N relative to Spire between 10 and 30 km impact height. The C2 and Spire occultations are collocated within 100 km and 3 hours of each other. C2 BA are approximately 0.15% larger than Spire BA. The N biases are much smaller, averaging about 0.02%. Fig.18 illustrates the geographic distribution of these biases at 20 km impact height, computed from 5° latitude-longitude binned values of C2 and Spire. Positive biases of C2 BA vs. Spire exist everywhere, but there are pronounced maxima between 40-45°NS. The overall biases in N are noticeably smaller everywhere, but there are also pronounced maxima between 40-45°NS. These maxima are caused in large part by sampling differences between C2 and Spire, mostly between 42.5° and 45° NS. Misleading values of biases can occur if the observations are not randomly distributed and there is a variation of the observation values with latitude or longitude. We looked at the counts and values of BA and N from C2 and Spire in 0.1° latitude bands between 42.5°-45° NS and found that the values of BA and N were similar in C2 and Spire, with both decreasing toward higher latitudes. However, the counts for C2 were much less than the counts of Spire in this band. Thus there are many more Spire observations with low BA and N compared to C2, and the bin averages of C2 are much larger than those of Spire.

The BA and N biases of C2 relative to Spire in Figs. 17 and 18 raise two questions: (1) Why are C2 BA positively biased <u>relative</u> to Spire, and (2) why are the N biases smaller than the BA biases, when the refractivities are computed from the BA? These questions are discussed in the next section.

Fig. 17: Biases of C2 BA (black) and N (blue) relative to Spire between 10 and 30 km MSL altitude for ROMEX period. The C2 and Spire occultations are collocated within 100 km and 3 hours of each other. Biases are normalized by the sample mean of ERA5.

Fig. 18: Mean differences in % of C2 and Spire BA (top) and N (bottom) at 20 km impact height, computed in  $5^{\circ}x5^{\circ}$  latitude-longitude bins and averaged over all days of ROMEX. The range of color scale is  $\pm$ -0.7% in both figures.

#### 5.2 Causes of C2 positive biases

791 792 793

797

799

821

The small positive BA biases of C2 relative to Spire and other ROMEX missions between 10 and 30 km result from their different orbit configurations around the nonspherical Earth. Because Earth is a spheroid, the local radius of curvature Rc varies with the latitude and azimuth angle of the RO occultation plane, except at the poles where it is constant in all directions. Azimuth angles are defined relative to the N-S direction (0° or +/-180° for occultation planes oriented N-S, and +/-90° for E-W). Therefore, for RO satellites with different orbital inclinations, the average R<sub>c</sub> differs, resulting in differences in bending angles at a given impact height. This variation of R<sub>c</sub> may be called the anisotropy of Earth's curvature and it has two effects on the BA, the azimuth effect and the sideways sliding effect. C2 is in a low-inclination orbit (24°), with all of its observations predominantly oriented within ±45° of the east-west (E-W) direction (Fig. 19a). In contrast, other ROMEX satellites (e.g. Spire and Yunyao) are in mostly highinclination (polar) orbits, with globally distributed observations and occultation planes generally oriented within ±45° of the north-south (N-S) direction (Fig. 19b,c) These differences in RO observing geometry, when combined with Earth's oblateness, result in systematic differences in bending angles as functions of impact height and altitude, thus introducing challenges when comparing RO data from missions with different orbital inclinations. However, the azimuth effect does not pose a problem for RO data assimilation because typically the 1D forward model already accounts for differences in azimuth angles through the variation in R<sub>c</sub>, ensuring that the modeled BA remains consistent with the RO observations in this respect.

**Deleted:** located within ±45° latitude and occultation planes predominantly oriented in an east-west (E-W) direction

Deleted: aligned in a

Fig. 19: Frequency distribution of azimuth angles for C2 (a), Spire (b) and Yunyao (c).

# 5.2.1 Representativeness differences due to azimuth angles of the occultation planes

832

834

\$42 843 844

850

852 853

854

855

856

The largest part of the C2 positive BA bias relative to Spire is explained by their different occultation plane azimuth angles, which result in representativeness differences (the azimuth effect). Occultation planes oriented E-W (as in most C2 occultations) have larger R<sub>c</sub> and azimuth angles than those oriented N-S (as in most Spire occultations) and the effect is largest at the Equator and zero at the poles (Fig. 20). Negative and positive values have the same effect, so only the absolute value of the azimuth angle is shown in Fig. 20. The variations of azimuth angle affect BA, but not N, which explains the overall smaller N biases in Figs. 17 and 18. If two atmospheres have the same N(z) but different R<sub>c</sub>, a ray with the same impact height traveling through the atmosphere with larger R<sub>c</sub> will accumulate a slightly larger bending angle, due to traversing a slightly longer path within an atmospheric shell, by a factor of  $\sqrt{R_c}$ . Although this effect is small, it can still cause a difference up to about 0.3% in the bending angles measured at the same impact height at the equator between azimuth angles in the N-S and E-W directions (the % difference in the square root of the R<sub>c</sub> associated with the two azimuth angles). However, because the Abel inversion uses the bending angle as a function of impact parameter, which inherently accounts for variations in Rc, it will recover the same N(z) from two different BA profiles.

Fig. 20: Variation of  $R_c$  with latitude (x-axis) and azimuth angle of occultation plane (y-axis). Note that  $R_c$  increases with latitude and the variation of  $R_c$  is larger at low latitudes compared to high latitudes.

In general, direct comparisons of BA from different RO missions are not physically meaningful unless the effect of azimuth angle is accounted for, typically through a model-based double differencing (DD) correction. In a presentation to IROWG-7 in September 2019, Bill Schreiner presented early results that showed a positive C2 bias of 0.1-0.2% relative to a combined dataset of MetOp and Kompsat-5 (Schreiner et al. 2019). This bias was reduced to nearly zero by DD using the ECMWF operational model. In DD the mean difference between two RO datasets is corrected by a reference model evaluated at each of the data sets (Tradowsky et al. 2017; Gilpin et al. 2019). For example, the C2-Spire bias shown in Fig. 17 is corrected using ERA5 by

$$\begin{aligned} \text{C2-Spire (DD)} &= [\text{C2-ERA5(C2)}] - [(\text{Spire-ERA5(Spire})] \\ &= \text{C2-Spire} - [\text{ERA5(C2)-ERA5(Spire})]. \end{aligned}$$

DD accounts for differences in the two data sets associated with other sampling differences such as temporal and spatial location differences, as well as those due to different azimuth angles and  $R_{\text{c}}$ . Fig. 21 shows that DD using ERA5 reduces the C2-Spire BA biases to an average of about 0.02% between 10 and 30 km impact height.

Fig. 21: <u>Biases of C2-Spire BA</u> before double differencing (black) and after double differencing (red). C2 and Spire are collocated within 100 km and 3 hours. Biases are normalized by the sample mean of ERA5.

# 5.2.2 RO retrieval biases related to the sideways sliding of the tangent point

In RO data retrieval, a single reference sphere, defined by a fixed center and radius of curvature anchored at the occultation point, is typically used to approximate the Earth's

surface throughout the entire RO profile. However, as the tangent point drifts horizontally, this reference sphere no longer accurately represents the local geometry of the Earth's ellipsoidal surface. As a result, rays that travel at certain heights above the true surface are mapped to different heights relative to the fixed reference sphere defined at the occultation point, thus contributing to observed positive C2-Spire biases. This effect is strongest in the tropics, where the difference between the radii of curvature along and across the ray path is greatest (Fig. 22), and negligible at the poles, where two radii of curvature are equal. This phenomenon was first explained in detail by Aparicio (2024). Due to the different distributions of azimuth angles of the occultation planes, the effect of sideways sliding of the tangent point, on average, results in positive biases in BA and N observations for satellites in low-inclination orbits such as C2 and negative biases in BA and N for satellites in high-inclination orbits such as Spire and the other ROMEX satellites. This effect, which has been ignored by all processing centers until recently, can be corrected by adjusting the impact heights by a correction termed the sideways sliding correction. This correction is simply the difference between local radius of curvature at the occultation point (within the occultation plane) and the distance from the center of sphericity to the reference ellipsoid at the estimated ray tangent point (which differs from the occultation point). Assigning the retrieved BA to an adjusted impact height is effectively equivalent to modifying the BA for a given impact height. Consequently, this adjustment further influences the refractivity as a function of altitude through the subsequent Abel inversion.

892

895

907 Fig. 22: Difference in radius of curvature ( $dR_c$  in km across minus along) ray path as a function of latitude (x-axis) and occultation plane azimuth angles (y-axis).

The magnitude of the correction varies with impact height depending on how the nominal location or point of an occultation (termed occultation point by UCAR and georeferencing by EUMETSAT) is defined (Weiss et al., 2025). UCAR defines the occultation point as where the L1 excess phase exceeds 500 m, which is typically in the lower troposphere. EUMETSAT defines it as the location where the straight line between the transmitter and receiver touches the ellipsoid (straight line tangent altitude SLTA or height of straight line HSL equals 0), which is in the upper troposphere-lower stratosphere (UTLS). The sideways sliding correction is smallest where the tangent point of the occultation is close to the occultation point. Therefore, for UCAR-processed data the correction is smallest in the troposphere, while for the EUMETSAT-processed data the correction is smallest in the UTLS (Marquardt, 2024, personal communication). When the correction is applied, the effect of different definitions of occultation point is largely eliminated (Sokolovskiy 2025, personal communication).

The effect of the sideways sliding correction to the C2 and Spire data processed by UCAR and the resulting C2-Spire BA and N biases are shown in Fig. 23. In contrast to the azimuth effect, the sideways sliding affects both the BA and the N biases. The reduction is smallest in the lower troposphere, because of the definition of the occultation point in the UCAR data. In the 20 to 40 km layer the correction reduces the C2 positive biases by up to 0.05%.

Deleted: B

Deleted: at 10 km

Fig. 23: Bias of C2 BA (black) and N (blue) relative to Spire for UCAR standard (solid) and sideways sliding-corrected data (dashed). C2 and Spire data for this comparison are collocated within 300 km and 3 hours. Biases are normalized by the sample mean of ERA5.

The magnitude of the sideways sliding effect depends on the antenna off-boresight angle. Small off-boresight angles (near zero) correspond to occultations with small sideways sliding; large off-boresight angles correspond to those with larger sideways sliding.

# 6 Summary and Conclusions

The Radio Occultation Modeling EXperiment (ROMEX) is an international collaboration to test the impact of varying numbers of radio occultation (RO) profiles in operational numerical weather prediction (NWP) models. An average of 35,000 RO profiles per day from 13 different RO missions from the United States, Europe, and China are being used in NWP models at major international centers to study how different numbers of RO profiles affect the analyses and forecasts. This paper evaluates the characteristics of the ROMEX data (bending angles and refractivities) processed by UCAR, with emphasis on the three largest datasets, COSMIC-2, Spire, and Yunyao.

ROMEX uncertainties (random error statistics) are estimated by the three-cornered hat (3CH) method, using short-term forecasts from the ERA5 and JRA-3Q reanalyses as

ancillary datasets. Biases are estimated by comparing the RO observations to models (ERA5 and ECMWF operational short-range forecasts) and to each other.

Overall, the statistical properties of the diverse ROMEX data after quality control are similar and suitable for NWP and other applications. The average relative (normalized) uncertainty variations in the vertical are similar, which supports the use of a common error model in variational data assimilation for all data sets. The biases are generally small (less than 0.15%) between 10 and 30 km which supports the use of RO data in NWP models as unbiased anchor observations. The average penetration depths (lowest height above surface retrieved in the data) are similar for most of the datasets, with more than 80% of the profiles reaching heights of 2 km or lower and 50% reaching 1 km or lower.

We evaluate in detail COSMIC-2, Spire, and Yunyao, which together comprise 78% of the ROMEX data. We compare the vertical and horizontal (global) variations of the bias and uncertainty statistics of these three datasets. The 3CH uncertainties of the datasets are similar. The biases with respect to each other and to models show small variations in the layer between about 8 and 35 km of approximately +/- 0.15%, which is important for climate studies and may be important for NWP when large numbers of RO are assimilated. This layer is often called the core region, golden zone, or sweet spot for assimilation in NWP models because the uncertainties and biases are smallest in this layer and are given the most weight in the data assimilation.

In some comparisons, COSMIC-2 (C2) shows a small positive bias of approximately 0.15% compared to Spire and Yunyao when the data are collocated. This bias is shown to be mostly a representativeness difference and is a result of their different orbits. C2 satellites are in low-inclination (equatorial) orbits, and Spire and Yunyao (and the other ROMEX data) are mostly in high-inclination (polar) orbits. These different orbits create two sources of biases.

The first source of the biases associated with the different orbits is different azimuth angles on the average, which account for about 0.1% positive bias for C2. This azimuth effect is a representativeness difference and not related to an intrinsic bias in the instrumentation or the processing. It can be reduced to near zero by double differencing using a model.

The second source is the horizontal sliding of the RO tangent point, which leads to a height difference between its position relative to the Earth's ellipsoid surface and the reference sphere. This difference results in a positive bias of less than 0.05% in the UCAR-processed C2 bending angle (BA) and refractivity (N) observations in the stratosphere compared to those of the polar orbiters. The *sideways sliding effect* can be easily corrected in the processing of the RO data by applying a correction to the impact height.

Future papers from the modeling centers will report on the impact of the ROMEX data on NWP model forecasts.

Deleted: n apparent,

Deleted: apparent

Deleted: rather than a true bias

Deleted: apparent

Deleted: apparent

1014 1015 1016 1017 1018 1019 1020

1029 1030

1032

1034 1035

1047 1048

1050 References

1052 1053

Code and data availability. The ROMEX data processed by EUMETSAT are available free of charge through ROM SAF under the ROMEX terms and conditions. Further information is available at https://irowg.org/ro-modeling-experiment-romex/ . The ROMEX data processed by UCAR are available from UCAR under the ROMEX terms and conditions. ERA5 data are available from the ECMWF data catalogue at https://www.ecmwf.int/en/forecasts/datasets/browse-reanalysis-datasets . JRA-3Q data are available from the Japan Meteorological Agency through the Data Integration and Analysis System (DIAS) at https://doi.org/10.20783/DIAS.645. The source code for these calculations and test datasets are available on request from the corresponding author.

Author contributions. Anthes developed the idea of ROMEX and the research plan for this study, guided the research throughout the project, and prepared the manuscript. Sjoberg developed the 3CH code, carried out many of the calculations, and contributed original ideas throughout the project. Starr assisted by carrying out many of the calculations and preparing the figures. Zeng led the theoretical work on the radio occultation biases. All co-authors contributed to the interpretation of the results and preparation and review of the manuscript.

Competing interests. None of the authors has any competing interests.

Acknowledgments. We thank the many people who have contributed to ROMEX and to this study. The COSMIC Data Analysis and Archive Center (CDAAC) team, Jan Weiss, Maggie Sleziak, Valentina Petroni, and Hannah Veitel, processed the ROMEX data and created several of the figures. Members of the ROMEX steering committee, Christian Marquardt, Hui Shao, and Ben Ruston, provided overall scientific leadership and technical expertise to ROMEX. Sergey Sokolovskiy and Stig Syndergaard provided valuable contributions to the understanding of biases. We thank all the ROMEX data providers and NWP centers for their essential contributions to ROMEX, much of it voluntary and without extra compensation. Support for this work was provided through NSF Grant 2054356, NASA Grant C22K0658, and NOAA Cooperative Agreement NA23OAR4310383B. We thank Eric DeWeaver (NSF), Will McCarty (NASA), and Natalie Laudier (NOAA) for their support.

Anthes, R.A., Rocken, C. and Kuo Y.-H.: Applications of COSMIC to meteorology and climate. Special issue of Terrestrial, Atmospheric and Oceanic Sciences (TAO), 11, 115-156. https://n2t.org/ark:/85065/d7fn16ch, 2000.

Anthes, R.A. and Rieckh, T.: Estimating observation and model error variances using multiple data sets. Atmos. Meas. Tech., 11, 4239-4260, 2018. https://doi.org/10.5194/amt-11-4239-2018, 2018.

Deleted: and UCAR

Deleted: also

Deleted: the apparent COSMIC-2

- Anthes, R., Sjoberg, J., Feng, X. and Syndergaard, S.: Comparison of COSMIC and COSMIC-2 Radio Occultation Refractivity and Bending Angle Uncertainties with
- Latitude in August 2006 and 2021. Atmosphere 13, 5.
- https://doi.org/10.3390/atmos13050790, 2022.

- Anthes, R.A., Marquardt, C., Ruston, B. and Shao, H.: Radio Occultation Modeling
   Experiment (ROMEX)-Determining the impact of radio occultation observations on
   numerical weather prediction. Bull. Am. Meteorol. Soc. 105,
- https://doi.org/10.1175/BAMS-D-23-0326.1, 2024.

Ao, C. O., D. E. Waliser, S. K. Chan, J.-L. Li, B. Tian, F. Xie, and A. J. Mannucci: Planetary boundary layer heights from GPS radio occultation refractivity and humidity profiles, J. Geophys. Res., 117, D16117, doi:10.1029/2012JD017598, 2012.

Aparicio, J. M. and Laroche, S.: Estimation of the added value of the absolute calibration of GPS radio occultation data for numerical weather prediction, Mon. Wea. Rev., 143, 1259–1274, https://doi.org/10.1175/2007MWR1951.1, 2015.

- Aparicio, J.M.: Bias derived from cross-track sliding of the occultation plane.
- Presentation to informal RO working group 15 November 2024 (personal communication), 2024.

Bonnedal , M., Christensen, J., Carlström, A.. and Berg, A.: Metop-GRAS in-orbit instrument performance. GPS Solut 14, 109–120, <a href="https://doi.org/10.1007/s10291-009-0142-3">https://doi.org/10.1007/s10291-009-0142-3</a>, 2010.

Bowler, N.: Personal communication in presentation at virtual meeting of ROMEX investigators 22 May 2024, 2024.

Bowler, N.: Revised GNSS-RO observation uncertainties in the Met Office NWP system. Q J R Meteorol Soc. 2020;146:2274–2296. https://doi.org/10.1002/qj.3791, 2020.

Caldeira, M.C.O., Caldeira, C.R.T., Cereja, S.S.A., Alves, D.B.M. and C.R. Aguiar, C.R.: Evaluation of the GNSS positioning performance under influence of the ionospheric scintillation. Bulletin of Geodetic Sciences. 26(3): e2020014. https://doi.org/10.1590/s1982-21702020000300014.2020.

Chang, L. C., Liu, H., Miyoshi, Y., Chen, C.-H., Chang, F.-Y., Lin, C.-H., Liu, J.Y. and
Sun, Y.-Y.: Structure and origins of the Weddell Sea Anomaly from tidal and planetary
wave signatures in FORMOSAT-3/COSMIC observations and GAIA GCM simulations.
J. Geophys. Res. Space Physics, 120, 1325–1340,

https://doi.org/10.1002/2014JA020752, 2015.

- Chen, Y.-J.; Hong, J.-S.; Chen, W.-J.: Impact of Assimilating FORMOSAT 7/COSMIC-2
   Radio Occultation Data on Typhoon Prediction Using a Regional Model. *Atmosphere*,
   1879. https://doi.org/10.3390/atmos13111879, 2022.
- Cheng, Y.; Fernhui, Li; Naifeng, F. and Peng, G.: Yunyao meteorological constellation and products introduction. Presentation at the 1<sup>st</sup> ROMEX Workshop April 17, 2024 at EUMETSAT, Darmstadt, Germany. Available at <a href="https://irowg.org/romex-1/">https://irowg.org/romex-1/</a>, 2024.
   Cheng, Y.: The improvement of Yunyao RO data. Quality and Development of Yunyao
   114 Constellations. Presentation at the 2<sup>nd</sup> ROMEX Workshop Feb. 27,2025 at
   1115 EUMETSAT, Darmstadt, Germany. Available at
   1116 https://www.eventsforce.net/romex2025, 2025.
  Cucurull, L., Anthes, R.A. and Tsao, L.-L.: Radio occultation observations as anchor
  observations in numerical weather prediction models and associated reduction of bias
  corrections in microwave and infrared satellite observations. J. Atmos. Oceanic
- Technol., 31, 20–32, https://doi.org/10.1175/JTECH-D-13-00059.1, 2014.
- Desroziers, G., Berre, L., Chapnik, B. and Poli, P.: Diagnosis of observation,
  background and analysis-error statistics in observation space. Quart. J. Roy. Meteorol.
  Soc., 131, 3385–3396. https://doi.org/10.1256/qj.05.108, 2005.
- 1127 EUMETSAT: Sentinel-6A GNSS-RO NTC Cal/Val Report. EUM/LEO-1128 JASC/REP/21/1243117, 1 NOV. 2022, 71 pp. Available at
- <u>https://user.eumetsat.int/s3/eup-strapi-</u>

1122

1149

- <u>media/Sentinel 6 A GNSS RO NTC Cal Val Report v1 E 3c966498f1.pdf,</u> 2022. 1131
- Eyre, J.: An introduction to GPS radio occultation and its use in numerical weather prediction. GRAS SAF Workshop on Applications of GPS radio occultation measurements, ECMWF, 16-18 June 2008. Available at
- https://www.ecmwf.int/sites/default/files/elibrary/2008/9342-introduction-gps-radiooccultation-and-its-use-numerical-weather-prediction.pdf, 2008.
- Feng, X., Xie, F., Ao, C. and Anthes, R.: Ducting and biases of GPS radio occultation bending angle and refractivity in the moist lower troposphere. J. Atmos. and Ocean. Technol., 37, 1013-1025, https://doi.org/10.1175/JTECH-D-19-0206.1, 2020.
  Gilpin, S., Rieckh, T., and Anthes, R.A.: Reducing representativeness and sampling
  errors in radio occultation-radiosonde comparisons. Atmos. Meas. Tech., 11, 2567–
  2582. https://doi.org/10.5194/amt-11-2567-2018, 2018.
  Gilpin, S., Anthes, R., and Sokolovskiy, S: Sensitivity of forward-modeled bending angles to vertical interpolation of refractivity for radio occultation data assimilation. Mon. Wea. Rev., 147, 148 269-289, https://doi.org/10.1175/MWR-D-18-0223.1, 2019.

Gorbunov, M. E., Vorob'ev, V.V. and Lauritsen, K.B.: Fluctuations of refractivity as a 1151 systematic error source in radio occultations, Radio Sci., 50, 656-669, 1152 https://doi.org/10.1002/2014RS005639, 2015.

Harnisch, F., Healy, S.B., Bauer, P., and English, S.J.: Scaling of GNSS Radio 1154 Occultation impact with observation number using an Ensemble of Data Assimilations. 1155 1156 Mon. Wea. Rev., 141, 4395-4413, https://doi.org/10.1175/MWR-D-13-00098.1, 2013.

Healy, S.B.: Forecast impact experiment with a constellation of GPS radio occultation receivers. Atmos. Sci. Lett., 9, 111-118, https://doi.org/10.1002/asl.169, 2008.

Hersbach, H., and Coauthors: The ERA5 global reanalysis. Q. J. Roy. Meteorol. Soc., 146, 1999–2049, https://doi.org/10.1002/qj.3803, 2020.

Ho, S.-p.; X. Zhou, X. Shao, Y. Chen, X. Jing, and W. Miller: Using the Commercial GNSS RO Spire Data in the Neutral Atmosphere for Climate and Weather Prediction Studies. Remote Sens., 15, 4836. https://doi.org/10.3390/rs15194836, 2023.

Ho, S.-P., Xi, S., Chen, Y., Zhou, J., Gu, G., Miller, W. and Jing, X.: Lessons Learned from the Preparation and Evaluation of Multiple GNSS RO Data for the ROMEX from NOAA/STAR. Presentation at the COSMIC/JCSDA Workshop and IROWG-10, Boulder, Colorado, 12-18 September 2024. Available at:

https://www.cosmic.ucar.edu/events/cosmic-jcsda-workshop-irowg-10/agenda, 2024.

1174 Ho, S.-P., Xi, S., Chen, Y., Zhou, J. and Miller, W.: Advances in ROMEX data processing and evaluation: Lessons from NOAA STAR. Presentation at the 2<sup>nd</sup> ROMEX 1175 1176 Workshop February 27, 2025 at EUMETSAT, Darmstadt, Germany. Available at 1177 https://www.eventsforce.net/romex2025, 2025.

Kosaka, Y., Kobayashi, S., Harada, Y., Kobayashi, C., Naoe, H., Yoshimoto, K., 1180 Harada, M., Goto, N., Chiba, J., Miyaoka, K., Sekiguchi, R., Deushi, M., Kamahori, H., 1181 Nakaegawa, T., Tanaka, T. Y., Tokuhiro, T., Sato, Y., Matsushita, Y. and Onogi, K: The 1182 JRA-3Q reanalysis. J. Meteor. Soc. Japan, 102, 49-109,

https://doi.org/10.2151/jmsj.2024-004, 2024.

1157 1158

1162

1164

1168

1178

Kuo, Y.-H., Wee, T.-K. Sokolovskiy, S., Rocken, C., Schreiner, W., Hunt, D. and Anthes, 1186 R.A.: Inversion and error estimation of GPS radio occultation data. J. Meteor. Soc. Japan special issue, 84, No. 1B, 507-531. https://doi.org/10.2151/jmsj.2004.507, 2004. 1187 1188

Kursinski, E. R., Hajj, G. A., Hardy, K. R., Schofield, J. T., and Linfield, R., : Observing 1190 Earth's atmosphere with radio occultation measurements, J. Geophys. Res., 102, 23429-23465. https://doi.org/10.1029/97JD01569, 1997. 1191 1192

Leroy, S. S., Anderson, J. G., and Dykema, J. A.: Climate benchmarking 1194 using GNSS occultation, in: Atmosphere and Climate: Studies by Occultation Methods, edited by: Foelsche, U., Kirchengast, G., and Steiner, A., 287-301, Springer-Verlag 1195

Berlin Heidelberg. https://doi.org/10.1007/3-540-34121-8 24, 2006.

Marquardt, C., vonEngeln, A., Alemany, F.M., Morew, N., Notarpietro, R., Paolella, S.,
 Padovan, S., Boscán, V.R. and Sancho, F.: The ROMEX Core Data Set-Statistics,
 Reprocessing, and Lessons Learned. Presentation at the COSMIC/JCSDA Workshop
 and IROWG-10, Boulder, Colorado, 12-18 September 2024. Available at:
 <a href="https://www.cosmic.ucar.edu/events/cosmic-jcsda-workshop-irowg-10/agenda">https://www.cosmic.ucar.edu/events/cosmic-jcsda-workshop-irowg-10/agenda</a>, 2024.

Marquardt, C.. R. Notarpietro, S. Paolella and A. von Engeln. Errors in Processing. Presentation at the 2<sup>nd</sup> ROMEX Workshop February 25-27, 2025 at EUMETSAT, Darmstadt, Germany. Available at <a href="https://www.eventsforce.net/romex2025">https://www.eventsforce.net/romex2025</a>, 2025.

Melbourne, W. G., and Coauthors: The application of spaceborne GPS to atmospheric limb sounding and global change monitoring. JPL Publ. Tech. Rep. 94-18, 159 pp., <a href="https://ntrs.nasa.gov/citations/19960008694">https://ntrs.nasa.gov/citations/19960008694</a>, 1994.

McNally, A.P.: On the sensitivity of a 4D-Var analysis system to satellite observations located at different times within the assimilation window. Quart. J. Roy. Meteor. Soc., 145, 2806-2816. <a href="https://doi.org/10.1002/qj.3596">https://doi.org/10.1002/qj.3596</a>, 2019.

Nielsen, J.K, Gleisner, H., Syndergaard, S. and Lauritsen, K.B.: Estimation of refractivity uncertainties and vertical error correlations in collocated radio occultations, radiosondes, and model forecasts. Atmos. Meas. Tech., 15, 6243–6256. https://doi.org/10.5194/amt-15-6243-2022, 2022.

O'Carroll, A. G., Eyre, J. R., and Saunders, R. S.: Three-way error analysis between AATSR, AMSR-E, and in situ sea surface temperature observations, J. Atmos. Ocean. Tech., 25, 1197–1207, https://doi.org/10.1175/2007JTECHO542.1, 2008.

Padovan, S., Von Engeln, A., Paolella, S., Yago, A., Galley, C.R., Notarpietro, R., Boscan, V.R., Sancho, F., Alemany, F., Morew, N. and Marquardt, C.: Observed impact of the GNSS clock data rate on Radio Occultation bending angles for Sentinel-6A and COSMIC-2. Atmos. Meas. Tech., https://doi.org/10.5194/amt-18-3217, 2025

Privé, N.C., Errico, R.M. and El Akkraoui, A.: Investigation of the potential saturation of information for the Global Navigation Satellite System radio occultation observations with an observing system simulation experiment. Mon. Wea. Rev., 150, 1293-1316, https://doi.org/10.1175/MWR-D-21-0230.1, 2022.

Qi, T.: Introduction to GNSS-RO and GNSS-R products of Tianmu-1 constellation.
Presentation at the 2<sup>nd</sup> ROMEX Workshop February 27, 2025 at EUMETSAT,
Darmstadt, Germany, Available at https://www.eyentsforce.net/romex2025, 2025.

Rieckh, T., Sjoberg, J. and Anthes, R.: The three-cornered hat method for estimating error variances if three or more atmospheric data sets-Part II: Evaluating recent radio

occultation and radiosonde observations, global model forecasts, and reanalyses. J. Atmos. and Ocean. Technol., 35, <a href="https://doi.org/10.1175/JTECH-D-20-0209.1">https://doi.org/10.1175/JTECH-D-20-0209.1</a>, 2021.

Ruston, B. and Healy, S.: Forecast Impact of FORMOSAT-7/COSMIC-2 GNSS Radio Occultation Measurements, Atmospheric Science Letters, 22, https://doi.org/https://doi.org/10.1002/asl.1019, 2020.

Semane, N., Anthes, R., Sjoberg, J., Healy, S. and Ruston, B: Comparison of Desroziers and Three-Cornered Hat Methods for Estimating COSMIC-2 Bending Angle Uncertainties. J. Atmos. and Ocean Tech., 39, 929-939. <a href="https://doi.org/10.1175/JTECH-D-21-0175.1">https://doi.org/10.1175/JTECH-D-21-0175.1</a>, 2022.

Schreiner, W., Sokolovskiy, S., Weiss, J., Braun, , J., Anthes, R., Kuo, Y.-H., Hunt, D., Zeng, , Z., Wee, T.-K., VanHove, T., Sjoberg, J. and Huelsing, H.: Performance Assessment and Requirement Verification of COSMIC-2 Neutral Atmospheric Radio Occultation Data. Presentation at IROWG-7, 19 September 2019, Helsingør, Denmark. Available at <a href="https://irowg.org/workshops/irowg-7/">https://irowg.org/workshops/irowg-7/</a>, 2019.

Schreiner, W.S., Weiss, J.P., Anthes, R.A., Braun, J., Chu, V., Fong, J., Hunt, D., Kuo, Y.-H., Meehan, T., Serafino, W., Sjoberg, J., Sokolovskiy, S., Talaat, E., Wee, T.-K., and Zeng., Z.: COSMIC-2 Radio Occultation Constellation-First Results. Geophys. Res. Lett, 47, e2019GL086841. https://doi.org/10.1029/2019GL086841, 2020.

Shao, H., Foelsche, U., Mannucci, A., Azeem, I., Bowler, N., Braun, J., Lonitz, K., Marquardt, C., Steiner, A., Ruston, B., and Panagiotis, V.: Advances in GNSS-Based Remote Sensing for Weather, Climate, and Space Weather: Missions, Applications, and Emerging Techniques. Bull. Amer. Meteor. Soc. Meeting Summary, <a href="https://doi.org/10.1175/BAMS-D-25-0138.1">https://doi.org/10.1175/BAMS-D-25-0138.1</a>, 2025

Sjoberg, J. P., Anthes, R.A., and Rieckh, T.: The three-cornered hat method for estimating error variances of three or more data sets-Part I: Overview and Evaluation. J. Atmos. and Ocean. Technol., 38, 555-572, <a href="https://doi.org/10.1175/JTECH-D-19-0217.1">https://doi.org/10.1175/JTECH-D-19-0217.1</a>, 2021.

Sokolovskiy, S.: Effect of superrefraction on inversions of radio occultation signals in the lower troposphere. Radio Sci., 38, 3, 1058, https://doi.org/10.1029/2002RS002728,2003.

Sokolovskiy, S.: Improvements, modifications, and alternative approaches in the processing of GPS radio occultation data. ECMWF/ EUMETSAT ROM SAF Workshop on Application of GPS Radio Occultation Measurements, Reading, UK, 16-18 June 2014. Available at <a href="https://www.ecmwf.int/en/learning/workshops-and-seminars/past-workshops/fifth-eumetsat-rom-saf-user-workshop-applications-gps-radio-occultation-measurements">https://www.ecmwf.int/en/learning/workshops-and-seminars/past-workshops/fifth-eumetsat-rom-saf-user-workshop-applications-gps-radio-occultation-measurements</a>, 2014.

Deleted: r
Deleted: o
Deleted: f
Deleted: r

Sokolovskiy, S.: Standard RO Inversions in the Neutral Atmosphere 2013 - 2020 (Processing Steps and Explanation of Data). <a href="https://www.cosmic.ucar.edu/what-we-do/data-processing-center/data">https://www.cosmic.ucar.edu/what-we-do/data-processing-center/data</a>, 2021.

 Sokolovskiy, S., Rocken, C., Schreiner, W. and Hunt, D.: On the uncertainty of radio occultation inversions in the lower troposphere, J. Geophys. Res., 115, D22111, <a href="https://doi.org/10.1029/2010JD014058">https://doi.org/10.1029/2010JD014058</a>, 2010.

Steiner, A. K., Ladstädter, F., Ao, C.O., Gleisner, H., Ho, S.-P., Hunt, D., Schmidt, T., Foelsche, U., Kirchengast, G., Kuo, Y.-Y., Lauritsen, K.B., Mannucci, A.J., Nielsen, J.K., Schreiner, W., Schwärz, M., Sokolovskiy, S., Syndergaard, S. and Wickert, J.: Consistency and structural uncertainty of multi-mission GPS radio occultation records. Atmos. Meas. Tech., 13, 2547–2575, https://doi.org/10.5194/amt-13-2547-2020, 2020.

Syndergaard S., Kuo, Y.-H., and Lohmann, M.S., 2006: Observation operators for the assimilation of occultation data into atmospheric models: A review. In: Foelsche U., Kirchengast G., Steiner A. (eds) Atmosphere and Climate. Springer, Berlin, Heidelberg, p. 205-224, DOI <a href="https://doi.org/10.1007/3-540-34121-8">https://doi.org/10.1007/3-540-34121-8</a> 18, 2006.

Syndergaard, S. and Lauritsen, K.: ROM SAF processing and new products. Presentation at the COSMIC/JCSDA Workshop and IROWG-10, Boulder, Colorado, 12-18 September 2024.

https://www.cosmic.ucar.edu/events/cosmic-jcsda-workshop-irowg-10/agenda, 2024.

Todling, R., Semane, N., Anthes, R. and Healy, S.: The Relationship between
 Desroziers and Three-Cornered Hat Methods Quart. J. Roy. Met. Soc., 148, 2942-2954.
 <a href="https://doi.org/10.1002/qj.4343">https://doi.org/10.1002/qj.4343</a>, 2022.

Tradowsky, J., Burrows, C., Healy, S., and Eyre, J.: A new method to correct radiosonde temperature biases using radio occultation data, J. Appl. Meteor. Clim., 56, 1643–1661, https://doi.org/10.1175/JAMC-D-16-0136.1, 2017.

Weiss, J., Veitel, H., Sokolovskiy, S., Zeng, Z., Anthes, R., Hunt, D., Petroni, V.,
 Sjoberg, J., Sleziak-Sallee, M. and VanHove, T.: Update on Processing and Analysis of
 ROMEX Data. Presentation at the 2<sup>nd</sup> ROMEX Workshop February 25-27, 2025 at
 EUMETSAT, Darmstadt, Germany. Available at
 https://www.eventsforce.net/romex2025, 2025.

Xie, F., Syndergaard, S., Kursinski, E.R. and Herman, B.: An approach for retrieving marine boundary layer refractivity from GPS occultation data in the presence of superrefraction. J. Atmos. and Oceanic Tech., 23, 1629-1644. https://doi.org/10.1175/JTECH1996.1, 2006.

Xu, X, Han, W., Wang, J., Gao, Z., Li, F., Cheng, Y. and Fu, N.: Quality Assessment of
 YUNYAO GNSS-RO Refractivity Data in the Neutral Atmosphere. Atmos. Meas. Tech.
 18, 1339-1353. <a href="https://doi.org/10.5194/amt-18-1339-2025">https://doi.org/10.5194/amt-18-1339-2025</a>, 2025.