# Peer review of "Evaluation of biases and uncertainties in ROMEX radio occultation observations"

_EGUsphere, 2025_

## Author Comment (AC1)

Reviewer #1
*Response in italics*

Paper Summary:

This paper analyses the random and bias error characteristics of data that was made available as part of the Radio Occultation Modeling Experiment (ROMEX). ROMEX is an agency and scientific community effort to quantify the benefits of assimilating increasing numbers of RO profiles for numerical weather prediction. In the three- month ROMEX experimental period, in excess of 35,000 profiles per day were made available to ROMEX participants who agreed to the terms and conditions of using ROMEX data. (This compares to less than 10,000 RO profiles per day that are available operationally.) The paper focuses on the three largest datasets from ROMEX: COSMIC-2 (C2), Spire, and Yunyao. The three-cornered-hat (3CH) method is used to determine the uncertainties of the C2, Spire, and Yunyao bending angles and refractivities. The analysis covers height dependence and geographic variations. Reanalyses are used as two corners of the 3CH method to estimate the uncertainty of the combined C2-Spire-Yunyao data set.

Review Summary:

The paper provides valuable scientific information on the ROMEX data, which is an unprecedented data set for RO and also involves new commercial sources of data such as Yunyao which has not been extensively analyzed in the past. This is the first time that such a large data set is evaluated for its error characteristics.

Understanding the error characteristics is very important for data assimilation and numerical weather prediction. Before publication, the paper should clarify certain aspects of the analysis and address questions as detailed below. After suitably addressing these aspects, the paper should be ready for publication.

Citation of the literature is generally appropriate, but with reliance on an unpublished document that could be given a DOI. The reference Aparicio 2024 is to a presentation that could be put online with a DOI.

The material is presented well in a logical and clear manner.

*General response: We thank the reviewer for these perceptive comments, which have led to improvements in the paper.*

Detailed Comments:

Line 107: a brief explanation of "excess phase data" should be provided so that the paper is accessible to a less specialized audience.

*Response: A radio wave is slowed as it passes through the atmosphere, which creates a doppler shift in the measured frequency of the wave and a delay in the phase of the wave as measured by the receiver. This delay is called the excess phase and is a fundamental observable in radio occultation (Kursinski et al. 1997). For details, please see The Radio Occultation Processing Package (ROPP) Pre-processor Module User Guide Version 11.3 (Section 3) [https://rom-saf.eumetsat.int/romsaf_ropp_ug_pp.pdf](https://rom-saf.eumetsat.int/romsaf_ropp_ug_pp.pdf) . The text has been modified in Section 1.1 of the revised paper, directing the reader to this reference.*

Lines 150-166: while the theory of the 3CH method is explained elsewhere, this brief summary does not fully

serve the paper. The paper compares different data sets to one another, and no data set is claimed to be "truth", so why is "truth" referred to here? Rather than truth, the authors might be referring to a reference data set for which biases and variances are determined relative to that reference. Wouldn't these equations still hold if one of the data sets is viewed as "reference" instead of truth? Or are these equations only valid if one of the data sets is actually "truth", which has zero bias and zero random error?

The term "bias" in this paper appears to refer to a bias between two data sets, and not between one data set and truth. What is meant by "bias" in the paper should be clarified.

Another way this brief introduction is not serving the paper is that the paper analyzes various subsets of data for which it becomes clear there is not a single bias applicable to all subsets. For example, bias appears to vary geographically, and the authors apply the analysis to the global data set which is not characterized by a single bias. Is the global bias expected to be the mean of the regional biases? Is equation (1) valid for a dataset that is characterized by multiple biases? The authors should clarify how the equations 1-3 apply to the data sets being analyzed in the paper.

*Response: In a discussion of the error model that is the basis for the derivation of the 3CH equations for error variances (Eq. 1 in our paper), O'Carroll et al. (2008) point out that the concept of "truth" is non-trivial. They indicate that for the derivation of the 3CH method error variances we only need to assume that "we have a consistent definition of the true value for comparison with each observation." In general, "truth" is a reference value for each triplet of data in the 3CH method rather than necessarily corresponding to the true value of the observation. Thus in Eq. 1 the use of the word "truth" is conceptual in nature. "Truth" does not need to be constant in space or time; it is only necessary that it be the same for each individual triplet because it cancels immediately upon taking the differences between the observations in each triplet. The validity of the 3CH estimates of error variances does not depend on any of the three observations in the triplet being "truth." A more complete discussion of the 3CH method, the meaning of "truth," the factors limiting the accuracy of the method, and comparison with other widely used methods for estimating uncertainties are described in Sjoberg et al. 2021. In summary, the 3CH method provides estimates of the random error statistics, not the biases, and hence does not depend on knowledge of "truth."*

*The term "bias" is used in two different ways in the paper. When discussing biases in general, we are referring to biases with respect to the true (always unknown) value at the scale (footprint) of the observation. When we refer to specific quantitative examples of biases we are referring to mean differences between one or more data sets with respect to a reference dataset, which may be a model or another observation dataset. We do not assume either of the two datasets is "truth." The difference in meanings should be clear from the context. We have clarified this in Section 1.3 of the revised paper.*

*Both the true (unknown) and estimated biases of a dataset vary with time and geographic location.*

*References*
*O'Carroll, A. G., J. R. Eyre, and R. W. Saunders, 2008: Three-way error analysis between AATSR, AMSR-E, and in situ sea surface temperature observations. J. Atmos. Oceanic Technol., 25, 1197–1207,* *https://doi.org/10.1175/2007JTECHO542.1.*

*Sjoberg, J. P., R.A. Anthes, and T. Rieckh, 2021: The three-cornered hat method for estimating error variances of three or more data sets-Part I: Overview and Evaluation. J. Atmos. and Ocean. Technol., 38, 555-572,* *https://doi.org/10.1175/JTECH-D-19-0217.1*

The following phrase is used on line 211: "but at the expense of fewer pairs in the sample and greater noise in the statistics." The description in lines 150-161 does not contain terms corresponding to this "statistical noise". For such a term to exist would require the concept of a sample mean as an estimate of the mean of a theoretical parent distribution. (Similarly for the parent standard deviation, etc.). The authors apparently are not concerned with statistical noise in their analyses, relying on large enough sample sizes to render the statistical noise negligible. The authors should make some reference to this implicit assumption in the paper.

*Response: The reviewer is correct. For our analyses of collocated datasets, the sample sizes far exceed the sample size of order 1000 suggested by Sjoberg et al. (2021) where statistical noise in 3CH estimates may be considered negligible. We have added this statement to the revised paper in Section 1.1.*

Line 270: provide some sense of what "impact height" is and how it relates to geometric height, for the less specialized reader.

Response: impact height is defined as the difference between the impact parameter and the Earth's local radius of curvature (Sokolovskiy et al. 2010). The impact parameter is a fundamental quantity used in RO technique, which refers to the distance between the ray asymptote to the local Earth's center. The impact parameter is constant along the ray path in a spherically symmetric atmosphere. Impact height is used in the retrieval of radio occultation observations and is related to the geometric height by the refractivity and the local radius of curvature of the Earth. Near the Earth's surface the impact height is approximately 2 km greater than the mean sea level altitude. The difference diminishes with height; by 10 km the difference is approximately 600 m and at 20 km the difference is approximately 125 m. The Radio Occultation Processing Package (ROPP) Pre-processor Module User Guide Version 11.3 (Section 3) https://rom-saf.eumetsat.int/romsaf_ropp_ug_pp.pdf describes the impact parameter. We have added a reference defining impact height in Section 2 of the revised paper.

Line 276-279: please clarify this sentence. It is hard to understand.

*Response: We have revised the sentence as follows and inserted the following text at the end of the preceding paragraph: "The magnitude of these differences (less than 0.15%) are much smaller than the 3CH uncertainty estimates, which are 1.5% or higher. However, they may have an impact on the comparison of bending angle biases, which are of the same order of magnitude between 10 and 30 km."*

Line 313-317: this paragraph and example should be better defined. There are numerous references that suggest RO can be used to advantage to improve predictions related to tropical cyclones (TC). What is meant by "resolve" here is not clear. RO inherently has relatively poor horizontal resolution per observation, which will not change with increasing numbers of observations. Even if RO cannot spatially "resolve" TC, increasing numbers of RO could improve predictions related to TC, such as intensification and track. This paragraph is not convincing regarding whether increasing numbers of RO have no benefit for TC.

*Response: We agree that this paragraph could be misleading. We did not mean to imply that RO observations have no benefit for TC prediction; to the contrary, RO observations have been shown in many studies to improve the predictions of TC (Chen et al. 2022 and references therein), as noted by the reviewer. We have revised this paragraph accordingly.*

*Chen, Y.-J.; Hong, J.-S.; Chen, W.-J., 2022: Impact of Assimilating FORMOSAT 7/COSMIC-2 Radio Occultation Data on Typhoon Prediction Using a Regional Model. Atmosphere, 13, 1879.* *https://doi.org/10.3390/atmos13111879*

Lines 324-325: if C2 does not exhibit the same count variation as Spire near the equator, it is worth commenting on why this might be case, if true. Both data sets sample the equatorial anomaly.

*Response: The Spire and COSMIC-2 data are based on different receivers, come from satellites with different orbits and have different signal-to-noise (SNR) ratios. These differences may be the reason why quality of the Spire observations is affected more by the Equatorial anomaly than the C2 observations, and hence relatively more Spire observations are rejected by the quality control. We added a short comment hypothesizing reasons for why the pattern does not exist in the C2 observation counts in the revised paper.*

Line 452: please provide the quantity (approximate) of operational data assimilated so that this statement can be provided in better context.

*Response: The number of RO profiles per day assimilated in the operational ECMWF model during the ROMEX period was between 7,000 to 7,500, or about 20-21% of the total ROMEX profiles. This number has been added to the revised paper.*

Lines 459-460: weren't the Yunyao data adjusted after the initial processing, so why does this artifact remain?

*Response: As stated in lines 137-139, we used the original data provided by the provider, not adjusted data that were provided later. Changes made in the processing of other ROMEX missions were likely made after the original data were submitted to EUMETSAT, but it would be difficult to obtain these data or decide on which revised datasets to use and which not to use. Thus, we evaluated data processed from the original data that were submitted to EUMETSAT.*

Line 502: Figure 7 is indeed impressive, but also somewhat puzzling. Whereas RO-RO comparisons have consistently shown growing uncertainties below 10 km, such uncertainty growth for the models is unexpected. For example, Figure 11 of Hersbach et al. (DOI 10.1002/qj.3803) for ERA5 temperature uncertainty does not appear to match what would occur with ERA5 bending angle uncertainty as indicated in Figure 7. While there is a modest increase of temperature uncertainty below ~8 km in the ERA5 paper, it does not increase dramatically towards the surface and is not much larger than uncertainty near 10 km. Please reconcile Figure 7 with Figure 11 of Hersbach et al.

*Response: While the ERA5 EDA spread can be interpreted as a measure for uncertainty, it is not the same thing as the random error variance. The main reason for the qualitative differences in the vertical variation of temperature spread in ERA5 in the troposphere from the vertical variation of refractivity or bending angle in Fig. 7 is related to the high variability and uncertainties in water vapor in the troposphere, which is an important part of the refractivity and bending angle observations. Thus, the fractional uncertainty of refractivity and bending angle is greater than that of temperature in the moist troposphere.*

Line 525: please clarify what BFRPRF in Figure 8 refers to. What are the dashed blue and green lines?

*Response: BFRPRF refers to the RO data that were processed by UCAR in its level-2 BUFR product, (described in lines 137-139); it refers to the three RO missions plotted in Fig. 8. We added this to the Fig. 8 caption. The blue and green lines are defined in the caption. We added a note in the caption that there are three estimates for the error variances of ERA5 and JRA-3Q, one for each RO mission. The differences are small (a good thing since they should not depend very much on the RO mission) and are not easily visible in this plot.*

Line 531: processing provenance is somewhat unclear. Were the Yunyao data used here reprocessed by EUMETSAT or UCAR?

*Response: As noted in lines 137-141, all the data analyzed in this paper were processed by UCAR. The excess phase data from Yunyao were first sent to EUMETSAT and then to UCAR, where they were processed to bending angle and refractivity.*

Line 583: please clarify what is the location of the ERA5 model and how is that determined. Is it a nearby grid point value? Isn't it straightforward to interpolate ERA5 values on a grid to an RO location, thus eliminating collocation error for all ERA5 comparisons?

*Response: We added more detail into how we calculated the geographic maps of the bias estimates between the RO datasets and ERA5 model data in Section 4.2 of the revised paper. An alternative way would be to interpolate ERA5 model values to each RO location, but we wanted to use the same method that we used to compute RO biases vs. another RO dataset as reference,, where collocation to each RO observation was not possible.*

Line 594: what is meant by short-range forecast of a reanalysis and why use that rather than the reanalysis value that is based on all contemporaneous data?

*Response: Short-range ERA5 forecasts refer to forecasts made by the ERA5 model initialized 6-18 hours before an observation time in order to minimize the dependence of the model data on the observations being evaluated and thus eliminate or reduce error correlations that affect the estimates of biases and uncertainties.*

Line 620: how is it known that above 30 km ERA5 biases are dominant?
Line 627: same comment – how is it known that ERA5 biases are dominant?

*Response: We cannot be sure whether the large mean differences we see in the ROMEX data vs. ERA5 are observation or model biases because there is no Truth at these levels. However, above 30 km the biases of all three independent RO datasets (C2, Spire, and Yunyao) show very similar bending angle biases compared to ERA5 (Fig. 10a). The mean refractivities of C2, Spire, and*

*Metop B/C all show very similar differences from the mean ERA5 refractivities (not shown in the paper). We have looked at several other estimates of different RO missions against different models, and the RO data from independent RO missions with different instruments, orbits, processing details, etc. all exhibit similar biases against the same model, but the models exhibit different biases with respect to each other. The consistency of all the RO data and the inconsistency in the models indicate likely biases in the models' bending angles and refractivities. These biases could be caused by biases in the model data or the forward model used to compute refractivities and bending angles from the model data (e.g. small errors in the $k_1$ coefficient in the refractivity equation). We have modified the text in Section 4.2 of the revised paper..*

Line 718: missing close of parentheses.

*Response: Corrected.*

Line 747: this suggests that the bias caused by Rc should not create model biases after assimilation because N does not exhibit the bias. The forward operator applied to bending angle should remove the bias. Please reconcile the small N bias with a bias in the model that assimilates BA. Note that Zhou et al. (DOI 10.1029/2024JD041295) detected temperature biases in C2 data. How do such biases arise if N is not biased due to Rc variations?

*Response: The azimuth effect, as defined in Section 5.2.1 of the paper, is a representativeness difference associated with variations in $R_c$ in different missions. We estimate that the BA difference of two occultations observed at the same location and time but along different azimuth angles, can be as large as 0.3%. This effect contributes to differences in direct comparisons of BA from different RO missions but won't raise an issue for BA data assimilation since the forward operator in the models accounts for the differences of $R_c$ in different missions.*

*While the azimuth effect does not cause the N bias, there are other factors, such as the one discussed in Section 5.2.2., that do, and these result in the small N biases found in our estimates. The relationship between N and dry temperature is complex and non-local, so there is no simple relationship between N biases and temperature biases. Furthermore, the temperature biases estimated in Zhou et al. and the N biases estimated in this paper are computed in different ways and have different QC applied to the data being evaluated.*

Lines 760-773: the DD examples used here relate to reducing collocation error. It's not immediately clear how DD reduces Rc error. Please provide more details regarding the algorithm that reduces Rc error. Include a discussion of how bending angle is computed from the model.

*Response: The DD algorithm is given by Eq. (4) in the original paper (now Eq. 3 in the revised paper). It corrects for all representativeness differences, and the azimuth effect (related to Rc error) is a representativeness difference, not an error. The DD method is described in detail by*

*Tradowsky et al. 2017 and Gilpin et al. 2019, as referenced in the paper. The model bending angles are computed using a 1D forward model that accounts for variations in $R_c$ in different missions, thereby accounting for the different azimuth angles and the azimuth effect. We added this information to Sections 1.1 and 5.2 of the revised paper.*

Lines 800-802: so the claim is being made here that the local radius of curvature computed for each occultation is only computed at one altitude, and not repeatedly as the ray path drifts?

*Response: Yes. The local radius of curvature is calculated at a single altitude for each occultation and does not change with the drift of the tangent points. Technically, it's possible to calculate the local radius of curvature at different altitudes, accounting for its variation with the drift of tangent points. However, this would introduce significant challenges in retrieving the refractivity profile. As far as we know, using a single local radius of curvature for the entire RO profile representation is the standard approach employed by all RO data processing centers.*

Line 808: What of negative azimuths?

*Response: The definition of azimuth may vary depending on the format of the RO data (e.g. BUFR or NetCDF) or data center from which it is downloaded. Herein, we use the atmPrf data processed by the CDAAC, where the azimuth angle of the occultation plane at tangent point is measured relative to North. Negative values indicate angles to the west of North.
Negative and positive values have the same effect, so only the absolute value of the azimuth angle is shown in Figs. 16 and 18. We have noted this in Section 5.2.1 of the revised paper.*

Line 848: this section should explicitly note whether bending angles or refractivity are being referred to.

*Response: These general conclusions refer to both bending angles and refractivity. We added this detail in the first paragraph of the Summary and Conclusions.*

*End of responses to Reviewer #1*

---

## Author Comment (AC2)

**Reviewer 2**

*Responses shown in italics.*

Short Summary:

The paper analyses data from the ROMEX experiment focusing on the three missions that contributed most data in the ROMEX period (September - November 2022). These are Spire, Yunyao, and COSMIC-2. Uncertainties and biases between datasets are addressed and sought understood. There are many interesting results, in particular global maps of uncertainties and biases, that I think deserve publication. However, I have concerns that the data that are analyzed are different from the data that have been provided to NWP users in the ROMEX experiment (if I understand it correctly). And I'm unsure if the data that are analysed are publicly available (or maybe not yet). I also have concerns about the correctness of some of the statements, which I elaborate in my specific comments below.

In my opinion, the paper needs major revision, with an update of some of the figures and a few more added, as well as modifications to many of the statements in the paper. Also more references needs to be included. The description of averaging method needs improvement. Generally the paper is well organized and the study is important.

*General response: We thank the reviewer for this careful and thorough review, which has contributed to the improvement of the paper. Some of the comments are complex and to fully respond to them is beyond the scope of this paper. However, we have tried our best to respond and would welcome further discussions with the reviewer.*

Specific comments:

L26: Maybe the COSMIC acronym should be spelt out. Or maybe it could be handled by saying "... referred to as COSMIC-2 (C2) ...".

*Response: COSMIC-2 is spelled out in the revised version: Constellation Observing System for Meteorology, Ionosphere and Climate-2.*

L33-34: "They are similar on the average in the region of overlap (45°S-45°N)". I don't think you can say this. According to Fig. 8, the Yunyao uncertainties are significantly larger (up to a factor of two) than Spire and C2 in the 10-15 km range. Is the sentence needed?

*Response: The differences between the uncertainties of the three missions over most of the atmosphere from about 5 to 55 km are much less than the magnitudes themselves (Fig. 8b), except for the 15-20 km layer where the Yunyao uncertainty is larger than the other two. However, this is an artifact of their initial processing and has since been*

*remedied. This is our qualitative assessment, and we prefer to keep it in the Abstract. The closeness of the uncertainties of these three very different independent missions is important because it means that the data can be treated similarly in model data assimilation.*

L37: "The assimilation of ROMEX data caused small degradations in biases in several NWP models". I suggest to remove this sentence from the abstract since it is not part of the study, and it is not mentioned anywhere else in the paper. It may give the impression that the biases studied in this paper was affecting the assimilation experiments, but I think it is now understood that they are likely not the cause of what NWP centers have found (presented at ROMEX workshops). I also question whether the data analysed in this study are similar enough to the data that were assimilated (see later comments).

*Response: We agree and have removed this sentence from the abstract.*

L43-45: Maybe this sentence in parenthesis is not needed in the abstract. It is explained later in the paper, and the detail is not needed here.

*Response: We have deleted the definition of the radius of curvature in the parentheses.*

L44: BA is not defined (but see my suggestion above of removing the sentence).

*Response: The phrase in the parentheses has been deleted.*

L68-69: "The ROMEX data became available at the European Organisation for the Exploitation of Meteorological Satellites (EUMETSAT) ...". I think you should add here "Radio Occultation Meteorology (ROM) Satellite Application Facility (SAF)". Then in line 123: "EUMETSAT Satellite Application Facility (SAF) on Radio Occultation Meteorology (ROM)" can become EUMETSAT ROM SAF.

*Response: Suggestion accepted.*

L87-88: "Spire and COSMIC-2 (C2) are relatively well known and have been widely studied". Please provide references.

*Response: We have now referenced Schreiner et al. (2020) for C2 and Bowler (2020) for Spire.*

L91: "... Germany 17-19 April 2024 (Cheng 2025)". Should it be "(Cheng 2024)"?

*Response: The workshop was held in 2024 but the reference to the workshop is 2025.*

L106-107: "The original (raw) data were downloaded ... by each data provider". Could you clarify which data providers you mean? It is not clear to me why data providers would need to download their own data.

*Response: The data providers are those who process the excess phase of each mission (e.g. COSMIC, Yunyao). We mean downloaded from the satellites and have added this clarification.*

L113-118: When was this done? Does this cover all ROMEX data or only a subset of them? Did EUMETSAT process these into refractivity? It appears that these data are different from the data described in the next paragraph (line 122 and forward), but it should be made much more clear up front why you mention these data and if it is those that are analysed in this study (or maybe it is only the UCAR part of them - needs to be more clear).

*Response: EUMETSAT relayed all the ROMEX excess phase data that it received from the providers to UCAR and STAR for further processing into bending angles, refractivities, and other products. This sentence has been modified for clarity. As described in L122-141 of the original paper, the data analyzed in this paper are the ROMEX data processed by UCAR. The bias and uncertainty statistics differ in some minor ways from the data processed by EUMETSAT. Most of the NWP modeling centers so far have used the EUMETSAT-processed data, which are being analyzed in detail by EUMETSAT. However, the UCAR-processed data also include a parameter that is related to the uncertainty of the observations in the troposphere, the local spectral width (LSW), which can be useful in data assimilation to estimate the RO error model of individual observation (Sjoberg et al. 2023) and a few centers are also using the UCAR-processed data.*

L116-118: "EUMETSAT, UCAR, and STAR processed the excess phase data into bending angles, refractivities, and other products, as described by Kuo et al. (2004) and Steiner et al. (2020)". But these references did not describe the processing of ROMEX data. Text needs modification to be clear. Perhaps mentioning EUMETSAT and NOAA STAR here just adds confusion.

*Response: These references describe in general terms how the processing of RO data is done, not the full description of the individual centers' processing algorithms, which are much too lengthy and detailed to be presented in this paper. The reader interested in these details may obtain them from the documentation provided by the processing center. We have added "as described generally by Kuo……" in the revised paper. We also added a reference to the details of the UCAR processing in the revised paper (Sokolovskiy, S.: Standard RO Inversions in the Neutral Atmosphere 2013 - 2020 (Processing Steps and Explanation of Data). 2020, Copyright 2021 UCAR 1 Standard RO Inversions in the Neutral Atmosphere 2013, 2021).*

*We think it is important to mention that two other centers also process the same ROMEX data, because these independently processed data may be useful for structural uncertainty and other studies.*

L130-135: I think you should move the citation in line 135 to line 130 as "Aparicio (2024)" instead of saying "Josep Aparicio (Environment Canada)". It is a problem that

this work is not publicly available. Did you check with Josep Aparicio if a publication is forthcoming?

*Response: We accepted this suggestion. We did check with Josep Aparicio, and he said he would include his work in a paper to be submitted to the ROMEX special issue of AMT. For now, it is a personal communication.*

L138-141: "... data that were originally provided to EUMETSAT and then processed by UCAR ...". Is this the data mentioned earlier (I think so, but it is not clear)? Are these data available anywhere (I could not find them)? In line 140 you say that UCAR-processed data and EUMETSAT-processed data are similar, but couldn't there be important differences larger than the differences between some missions? Three potential examples comes to mind after reading the whole paper:

1) Is there a different penetration for Metop in Fig. 4. than what was provided to NWP centres?
2) What is the influence of different definitions of the occultation point?
3) Could refractivity biases be significantly different at high altitudes due to different approaches in statistical optimization?

*Response: Yes, all the data analyzed in this paper are the data processed by UCAR, after receiving the excess phase data from EUMETSAT. We agree that some differences between the UCAR and EUMETSAT processing have been discovered recently and have qualified this statement. The refractivity biases at high levels (above 40 km) may be different due to the different statistical optimizations, but we do not discuss the high-level refractivities in this paper.*

*The UCAR-processed ROMEX data were just recently added to the ROM SAF ROMEX server, but have always been available from UCAR upon request.*

L139: "In sensitivity studies to investigate structural uncertainty (Steiner et al. 2020), we find ...". I think you should skip the citation as it is not directly related to the data analysed here. Or phrase things differently.

*Response: Although this is not a paper about comparing UCAR- and EUMETSAT-processed ROMEX data, we thought we should say something about what we have found so far about the differences between the two datasets. We have reworded the sentence to: "Performing structural uncertainty analyses similar to Steiner et al. (2020), in limited comparisons we find that the UCAR-processed data and the EUMETSAT-processed data are similar in most respects; examples are shown in the Supplement (S9)."*

L141: "... processed data are similar. Examples are shown in the Supplement.". Please provide the figure numbers?

*Response: We show examples of UCAR vs EUMETSAT processed data in S9 of the Supplement as noted in the revised sentence above.*

L144-146: "The penetration rate is defined as the percentage of successful occultations reaching different levels ...". Is it one number? or several numbers? It is not clearly defined here or anywhere else. Maybe it should not be called "rate".

*Response: We have changed the sentence to "The penetration depth is defined as the percentage of profiles of a sample of RO observations reaching different levels above the ground." It is a number (%) for each level of a sample of RO profiles. For example, a penetration depth of 80% at 2 km for a sample of RO profiles means that 80% of the profiles in this sample reach 2 km. It is a single number for each sample and each level.*

L160-166: "The root mean square error is ... RMS" Why mention it if it isn't used?

*Response: Many papers use RMS differences, and so we thought that was useful to distinguish RMS from STD. However, we agree it is unnecessary and have deleted it.*

L172: "RO biases are therefore assimilated in NWP models without bias corrections". Do you mean "RO observations are therefore assimilated ..."?

*Response: Yes, we have corrected the sentence.*

L180-183: "... commonly used verification charts ... often plotted together on a scale of -20% to +20%". Please provide references.

*Response: We have added Schreiner et al. (2020) and Ho et al. (2023) as examples.*

L184-189: Please provide specific references to support the statements here (I suppose from the ROMEX workshops).

*Response: Yes, the ROMEX workshops. We have noted this in the revised paper and provided a link to the IROWG website that has all the ROMEX workshop presentations: irowg.org/romex-events-meetings/.*

L194: "sometimes called the RO core region, golden zone, or sweet spot" Please provide references.

*Response: We don't know of any reference that uses all three of these terms. These are colloquial terms often used in the community and we define them here. We have revised the text slightly to "sometimes colloquially called….".*

L200-204: It would be good to mention here that the analyses and forecast data are forward modelled to BA and N.

*Response: This is an important point; we agree and have added this to the appropriate sentence and also described how errors in the forward model could contribute to the perceived model RO biases.*

L210: "Nielson" should be "Nielsen".

*Response: Corrected.*

L269: "We compare bias and uncertainty profiles...". I suggest to say instead: "We compare bending angle observations ...". It is only the BA observations that are compared on impact heights, not the refractivity.

*Response: Suggestion accepted.*

L269-272: Would it be more correct to say that the BA observations depend on five parameters: time, impact height, latitude, longitude, and azimuth? The dependency on azimuth is ignored in the comparisons, but not the dependency on the other four parameters.

*Response: Although the BA observations depend on these five parameters, only the small difference in azimuth angles of different missions make the plotting of BA vs. impact height not strictly physically meaningful (because of representativeness differences), so that is why it is mentioned here. We have modified the text to read "... in some of our results. The influence of the occultation plane's azimuth angle is not considered in these comparisons and will be further discussed in Section 5."*

L276: "The issues ... are much smaller than 0.15% ...". Issues are not quantitative (the representativeness differences are). Please modify text.

L278: "comparing RO missions on impact height ...". I suggest: "comparing RO missions ignoring the azimuth dependency ..."

L280: "However, they likely ...". Now 'they' refer only to issues, not to the representativeness differences. Please modify text.

*Response to previous three comments: We have rewritten this text and combined it with the previous paragraph:*

*"In some of our results, we compare bending angle bias and uncertainty profiles of the ROMEX missions as a function of impact height, which is related to the geometric height by the refractivity and local radius of curvature of the Earth (Sokolovskiy et al. 2010). The influence of the occultation plane's azimuth angle is not considered in these comparisons and will be further discussed in Section 5. These are representativeness differences and not differences in the quality of the retrievals. The magnitude of these differences (less than 0.15%) is much smaller than the 3CH uncertainty estimates, which are 1.5% or higher. However, they may have an impact on the comparison of bending angle biases, which are of the same order of magnitude between 10 and 30 km."*

L321: ... "in the 40-45°NS bands". I suggest to note here what NS means.

*Response: We have defined NS the first time it appears, i.e. 40-45°NS (40-45° north and 40-45° south).*

Figure 1: The first five figures span four pages, but are numbered 1a to 1e, with separate captions and without any direct relation. I suggest to number them 1 to 5. Similar could be said about Figure 10a and 10b,c. (They could be two separate figures).

*Response: Since these two figures are all related, we prefer to group them as done in the paper. This does not affect the length of the paper.*

Figure 1a: It could be mentioned in the figure caption that the underlying world map is for 12 UTC, but that it is not important. The x-axes could be labelled "local time [hr]".

*Response: We have added to the caption: "The x-axes are local time in hours."*

Figure 1e: This plot gives the impression that there are very few occultations at high latitudes for the polar missions. It would be more interesting to see the density per unit area (divide by cos(lat)), which I think would be more relevant.

*Response: We have added a panel to Fig. 1e showing the number of occultations per 10000 km$^2$ as a function of latitude. Indeed, this does present a different perspective.*

Figures in general: There is often information in the titles above figures which should be in the captions, e.g., "20 km" in Fig. 2 should probably be mentioned in the figure caption. Please check all figures and modify captions as needed.

*Response: We agree and have checked all figures and modified captions appropriately.*

L374: "Fig. 2 shows the daily BA profile counts (after UCAR CDAAC QC but before the 3CH QC)". What is the 3CH QC?

*Response: It is the QC (quality control) described in the first paragraph of Section 2.3. We have modified the text to say "after UCAR CDAAC QC but before the 3CH QC as described in Section 2.3."*

L433: "penetration rates are noticeably less for Metop-C (green), ...". And Metop-B too, it seems (I would say that the colors for Metop-B and Metop-C are olive green and/or dark cyan, but maybe there are more precise color names provided by the plotting software that could be used here). Why is the penetration so poor for Metop? Is it the same in the ROMEX core data that can be found at the ROMEX data server, or is this an artefact of the CDAAC processing?

*Response: The penetration depth profiles of Metop-B and Metop-C here are noticeably poorer than the other ROMEX missions. Although not shown here, they are also poorer than the EUMETSAT-processed Metop data. Thus, this is probably an artifact of the*

*CDAAC processing and is undergoing investigation. We have inserted a sentence to this effect in the revised paper.*

L436: "These results confirm that radio occultation is a useful method of obtaining global information on the planetary boundary layer." Please provide references (or maybe the sentence is not needed here).

*Response: We have added a reference here: Ao, C. O., D. E. Waliser, S. K. Chan, J.-L. Li, B. Tian, F. Xie, and A. J. Mannucci (2012), Planetary boundary layer heights from GPS radio occultation refractivity and humidity profiles, J. Geophys. Res., 117, D16117, doi:10.1029/2012JD017598.*

L448: "In this section we present an overview of the bias and uncertainty statistics of all the ROMEX data. Many additional detailed results are presented in the Supplement.". But the supplement only contains details about the three missions that are the focus in the next sections, not the rest of the ROMEX data that are discussed in this section (Section 3). Maybe the last sentence is not needed.

*Response: We have revised the last sentence to read: "Many additional figures showing statistics for the three largest ROMEX datasets are presented in the Supplement."*

L451-455: Say "MSL altitude" instead of "MSL height" (altitude is commonly used with reference to MSL). It would be good to mention here that it is the MSL altitude of the tangent points (I suppose).

*Response: Suggestion accepted. We have added that it is the altitude of the RO tangent points.*

L457: "... having the smallest uncertainties because of their more accurate clocks (Padovan et al. 2024).". Padovan et al. only looked at Sentinel-6 and COSMIC-2 and did not compare to other missions. Could the CDAAC processing also here have an influence?

*Response: We have added a reference on the more accurate Metop data because of their more accurate clocks (Bonnedal et al. 2010). We are not sure why the CDAAC reference is relevant here, since CDAAC starts with the excess phases and all the datasets being processed use consistent algorithms.*

Figure 5: Are all the missions processed by CDAAC with the same software version? If not, I suggest to add a table which shows processing versions and/or origin of the processing (e.g., if the low level processing was done by another data provider).

*Response: All the missions shown in Fig. 5 (as well as other figures) were processed with the same software version after the excess phase data were received from EUMETSAT. The excess phase data were obtained from EUMETSAT, which in turn received them from each data provider as described in Section 1.1.*

L458: "Fengyun-3 shows higher uncertainties between 10-30 km than the other missions. Yunyao has a peak in uncertainties between 10-15 km ...". Should it be "uncertainty" (singular)? I don't think it is correct to say "between X-Y km". Either say between X and Y km, or in the X-Y km range (or similar). Please modify all such occurrences throughout the paper.

*Response: There are multiple values of uncertainties that are plotted between 10 and 15 km so plural seems appropriate. We have changed the notation for layers from "between X-Y km" to "between X and Y km" throughout the paper.*

L468: "... ROMEX missions between 10-35 km (Fig. 6a)". This is the first reference to Fig. 6, but before that, it should be properly introduced in the text (not in a parenthesis). I noticed the same regarding Fig. 15 and Fig. 17.

*Response: We have changed the sentence to "ROMEX missions between 10 and 35 km, as shown in Fig. 6a." This should be sufficient to introduce Fig. 6.*

L475: "Fig. 6: Upper left (a) ...". Letters (a,b,c,d) are missing in the Figure. Please add letters to each of the four panels.

*Response: The letters have been added to Fig. 6.*

Figure 6: There is a confusing background grid with dots over all four panels that doesn't belong here. It needs to be removed. Same can be said of Fig. 4.

*Response: Dots have been removed in Figures 4 and 6.*

L486-487: Perhaps replace "in these latitudes" with "at these latitudes".

*Response: Suggestion accepted.*

L489: "figures show the importance of comparing different RO missions using spatial and temporal sampling as similar as possible". Why temporal? At first glance, I don't see any effect of different temporal samplings. Please clarify.

*Response: We agree that Fig. 6 shows the importance of similar spatial sampling and not temporal sampling and have removed "temporal" from the sentence.*

L503: "... the simple but effective error model used by ECMWF (Ruston and Healy 2022) is shown as a dashed line." I assume you mean "error model for RO observations". The general reader could misunderstand this to be the error of the ECMWF operational model, especially when one looks at the legend in Figure 7. I suggest to use a more precise indication than "ECMWF" in the legend to distinguish the meaning from that of "ERA5" and "JRA3Q".

*Response: We have revised this line to say "simple but effective RO observation error model used by ECMWF." We have noted that the label "ECMWF" in the legend refers to the ECMWF RO error model rather than the uncertainties of CSY and ERA5 derived from the 3CH method.*

L504: "Considering that it was developed many years ago, the agreement with the CSY data is remarkable.". I think this is an overstatement. The ECMWF-used error estimate is significantly larger at high altitudes. Please modify text.

*Response: We have modified the text to emphasize that the close agreement between the ECMWF RO error model and the CSY uncertainties is between 10 and 35 km.*

L533: "uncertainties and biases below 5 km are related to their cutoff of carrier phase data ..." This is the first mention of a bias in Yunyao data below 5 km, and it is not shown in Fig. 8. Should it be shown?

*Response: We have removed "and biases" from the sentence because Fig. 8 shows only the uncertainties. We do not investigate possible biases in the Yunyao data in this paper.*

L534: "... too early, as described by Sokolovskiy (2014) and noted by Marquardt et al. (2024).". Sokolovskiy (2014) did not describe Yunyao data, so I don't think you should cite it like this. In the conclusion of (Xu et al. 2025) it says: Larger biases are primarily observed in the lower troposphere, a phenomenon that has been extensively discussed in previous studies (Sokolovskiy et al., 2014; Xie et al., 2010). However, I think that is with reference to the more recent data version that Yunyao has processed. Are you sure that the much larger bias in the Yunyao data in your study can be attributed to the effects described by Sokolovskiy (2014)?

*Response: Thank you. As noted above, we did not look at the Yunyao biases. We have removed "biases" from the discussion of the Yunyao data in this sentence and only discuss uncertainties.*

L535: I think it should be (Xu et al. 2025).

*Response: Yes, we changed it to 2025.*

Figure 8: It is very difficult to distinguish solid, dash-dotted, and dashed of the same color. Given the curves in Fig. 7, would it make sense to show differences to these in Fig. 8? Like Fig. 6, this figure also has a confusing background grid that needs to be removed.

*Response: We prefer to show the error standard deviations of the three missions. The difficulty in distinguishing the three RO missions emphasizes the important point that all three missions are similar over most of the heights. We have added details about the similarity of the profiles in the Figure caption. We have deleted the background grid from Figs. 4, 6, and 8.*

L551: "Larger uncertainties occur over Asia and the Pacific". I see it in many other places in the tropics. Why single out Asia and the Pacific? It is difficult to see that the uncertainty is particularly larger there.

*Response: We agree and have deleted this sentence.*

L552-554: "... interesting regional feature is the maximum uncertainty over the Weddell Sea at 20 and 30 km, which may be related to the ionospheric Weddell Sea anomaly (Chang et al. 2015).". If it is related to ionospheric disturbance, then why would it not be seen at 50 km? Could there be other explanations? It is indeed very interesting!

*Response: The color scale of the uncertainty map is larger by more than an order of magnitude at 50 km than it is at 20 and 30 km; thus, the feature of interest visible at 30 and 40 km does not show up at 50 km.*

L564: "Fig. 9: Global distribution of 3CH uncertainties (%) for CSY BA at 3 km (a), 5 km (b), ...". Letters are missing in the figure panels.

*Response: Letters have been added.*

L570: "Larger versions of the panels are presented in the Supplement." Actually, the Fig. 9 and Fig. 12 maps are a bit more sharp (when zooming in) in the paper than in the Supplement in my pdf viewer (acrobat).

*Response: Yes, the figures in the Supplement are not quite as sharp as those in the paper because we had to compress the Supplement in order to meet the AMT size limit. However, we find all of the figures acceptably sharp to show the main features in the uncertainty distributions.*

L574-576: "The small negative impact of the ROMEX data on the biases of several NWP models has caused intensive study of possible causes of these small biases, including the possibility of small biases in the ROMEX data." Would a reference to 'this issue' (or similar) be possible here? (I assume that the paper, when it is in its final form, will be included in the special ROMEX issue).

*Response: We added a reference to the ROMEX workshops at the beginning of Section 4.2 where these NWP biases are mentioned. Yes, our paper will be included in the ROMEX special issue.*

L584-585: "... collocate an RO dataset with a model or another RO dataset, with advantages and limitations discussed in Section 2.". Could you briefly remind the reader what the advantages and limitations are? It is not totally clear to me.

*Response: "Section 2" is a typo. We describe the collocation of an RO dataset with either a model or another RO dataset in the last paragraph of Section 1.1. We have clarified*

*this in the second paragraph of Section 4.2 in the revised paper so that the reader can easily find the advantages and limitations of the two collocation methods.*

L586-587: "... the two RO datasets in latitude-longitude bins and compute the averages over each bin of the difference between the RO and reference data (e.g., ERA5).". This raises a number of questions:

1) How do you handle drifting tangent points? Can one part (say the lower part) be averaged within one bin, and another part (say the upper part) be averaged within another bin? Or is it at a reference location for the whole profile?

2) Is area weighting with a cosine factor taken into account when averaging over latitude?

3) Are sampling errors taken into account?

You need to describe your method in much more detail, possibly with equations, so that others can reproduce it. Gleisner et al. (2020) (https://doi.org/10.5194/amt-13-3081-2020) gives a nice description of how one can average in 5 degree latitude bins, including sampling error correction. I suppose your method is slightly different. Please describe it.

*Response: We have rewritten this paragraph and added more details of the second way (collocation within latitude-longitude bins and comparison with model data in the bin) in the revised paper. The tangent point of each profile is used to locate the RO observation in the appropriate bin at each level, and so one part of a profile can be in one bin and another part in another bin. There is no area weighting; this is simply an estimate of the mean differences between a sample of RO data and a reference dataset.*

L587: "This results in large samples and all RO data can be included.". Do you mean all three months? Please clarify.

*Response: This statement is true in general (all RO data in any sample can be included) including the three-month sample studied here.*

L602: "propagates into N after the Abel inversion". Since this is the first mention of the Abel inversion, perhaps a reference to the Abel transform would be good here for the general reader.

*Response: We added Kursinski (1997) as a reference.*

L617: "computed from 5x5° latitude-longitude bin averages over all bins and days of ROMEX." Do small bins at high latitudes contribute with the same weight as large bins at low latitudes?

*Response: We are comparing RO vs. ERA5 biases on a per-profile basis (i.e. each RO profile is included in the sample regardless of latitude). There is no latitude weighting*

*because this bias estimate is only for the entire RO dataset—it is not representative of a latitude-weighted global sample as used for climate studies. Please see also the response above to Comment L586-587.*

L618: "bases" should be "biases".

*Response: Corrected.*

L630-631: "Fig. 11 shows the mean differences between Yunyao and C2 from Spire between 10-40 km impact height.". I suggest rephrasing to something like: "Figure 11 shows Yunyao and C2 normalized biases relative to Spire between 10 and 40 km.".

*Response: Suggestion accepted.*

L634-637: "The bulge between 15 and 20 km is likely related to the relatively large horizontal sampling differences in the 5x5° latitude-longitude bins in a layer with large variations of atmospheric densities in the vicinity of the tropopause." I don't understand how that could be a reason. Please elaborate and please clearly describe how the results in Fig. 11 are obtained. Do they come from collocations? What does "2-day Mean" in the title above plots mean?

*Response: We have added a more detailed description of how the profiles and maps of the biases and uncertainties are obtained using the 5x5° bins, including the meaning of "2-day mean" near the beginning of Section 4.2 as discussed above. We have also added additional evidence for the cause of the "bulge" between 15 and 20 km being related to the systematic sampling differences in the 5x5° bins–the fact that the "bulge" does not exist when the Spire and C2 observations are collocated very closely (Fig. 13). Finally, we added the Yunyao minus Spire differences with Yunyao restricted to 45°NS in Fig. 11 for better comparison with the C2 differences.*

L651-652: "ERA5 biases may be of comparable or larger magnitude at all levels." I don't think ERA5 biases are this large at the lower levels. Would you agree?

*Response: This is an important point. The ERA5 (and other model) biases of BA and N are computed from a forward model using the model data (temperature, pressure, and water vapor). Biases may occur from systematic errors in the forward model as well as the model data. Both contribute to what we call "ERA5 biases." We do not think the ERA5 model data biases themselves are necessarily as large as these RO biases would indicate. We have clarified what we mean by ERA5 and other model biases in the revised paper (Section 1.1 L219-221) in the revised manuscript.*

L670-672: "The ROM SAF Matched Occultation page presents daily estimates of the biases of RO satellites compared to other RO satellites, with a collocation criteria of 300 km and 3 hours (https://rom-saf.eumetsat.int/monitoring/index.php )." I think the collocations are more precisely at https://rom-saf.eumetsat.int/monitoring/matched.php .

*Response: We agree and have used the more precise link in the revised paper.*

L672-674: "This monitoring site shows various combinations of mean and standard deviation of differences between BA and N from different satellites". I don't think the sentence makes sense (Combinations of mean and standard deviation? Combinations of differences between BA and N?). Please rephrase.

*Response: We have deleted "various combinations of" from the sentence.*

L674-676: Perhaps the paper should not be discussing details in a figure from someone else's website. Is it necessary?

*Response: We think this reference to C2 biases with other RO missions from an independent source such as the widely used ROM SAF monitoring site is an important source of independent information supporting the small positive biases of C2 BA, similar to other references included in this paragraph.*

L694: "Why are C2 BA positively biased to Spire ...". Maybe say "biased relative to".

*Response: Suggestion accepted.*

L711: "computed from 5x5° latitude-longitude bin averages over all bins ...". I think there may be a copy-paste mistake here. There are no averages over bins in Fig. 14, right?

*Response: Correct. There are averages within each bin and over all days of ROMEX. We have corrected the caption.*

L716: "The small positive biases of C2 relative to Spire and other ROMEX missions in BA". Suggestion: "The small positive BA biases of C2 relative to Spire and other ROMEX missions".

*Response: Suggestion accepted.*

L717: "... between 10 and 35 km result from their different orbits around the non-spherical Earth". I suggest to say "result from their different orbit configurations."

*Response: We have revised the sentence to say: "...result from their different orbit configurations around the non-spherical Earth." We think it is important to mention the role of the non-spherical Earth.*

L719: "... that best fits the Earth's surface curvature at a given location ...". Ending parenthesis is missing. I would rather say: "... that fits the Earth's surface curvature along the direction of signal propagation ...".

*Response: Suggestion accepted.*

L722-723: "The effect of this variation of Rc on the BA ...". Perhaps say "the azimuth" instead of "this" to be clear.

L723: "... may be called the anisotropy of Earth's curvature effect.". Do I understand correctly that this covers both the azimuth effect and the sideways sliding effect? Should it be plural (effects)?

*Response: We have rewritten the sentence in L722-723 as: "This variation of Rc may be called the anisotropy of Earth's curvature, and it has two effects on the BA, the azimuth effect and the sideways sliding effect."*

L724-725: "C2 is in a low-inclination orbit (24°), with all of its observations located within ±45° latitude and occultation planes predominantly oriented in an east-west (E-W) direction.". I'm not convinced of this. I suggest to include a figure showing the distribution of occultation plane azimuth angles for C2 and Spire.

L741: "... oriented E-W (as in most C2 occultations) ...". Please show it.

*Response: We have included a figure (new Fig. 15) showing the frequency distribution of azimuth angles for C2, Spire, and Yunyao (shown below).*

[Figure]

*Fig. 15 (new): Frequency distribution of azimuth angles for C2, Spire and Yunyao.*

L746: "larger Rc will accumulate a slightly larger bending angle, due to traversing a slightly longer path." My understanding is different: A signal is being affected by slightly more perpendicular gradients (due to the slightly larger curvature) on its path in the vicinity of the tangent point (vertical refractivity gradients are precisely perpendicular at the tangent point, but gradually less perpendicular away from the tangent point because of the Earth's curvature), thus accumulating more bending. The path length may in principle be the same. Would you agree with this?

*Response: Thanks for this question. The explanation in the paper "slightly longer path" was used for brevity.*

*Considering a thin atmospheric shell of thickness H ($H \ll Rc$) over a sphere of radius Rc. The GNSS signal can be approximated as a straight line through this layer. The horizontal half-path inside the layer is $L \approx \sqrt{2RcH}$. This is an effective length, which is proportional to $\sqrt{Rc}$. An increase $\Delta Rc$ results in a fractional change $\Delta L/L \approx (\Delta Rc/Rc)/2$. The angle between the perpendicular to the ray and local radius vector on exit from the layer is $\beta \approx L/Rc \approx \sqrt{2H/Rc}$. Accordingly, the perpendicular component of the unit vector (associated with the refractivity gradient, which is another contributor to the bending angle noted by the reviewer) is $n_\perp = \cos\beta \approx 1 - \beta^2/2 \approx 1 - H/Rc$. An increase $\Delta Rc$ causes a fractional increase $\Delta n_\perp/n_\perp \approx (\Delta Rc/Rc)(H/Rc)$.*

*Thus, both factors i) the increase of the effective propagation length L and ii) the increase of the perpendicular component of the refractivity gradient proportional to $n_\perp$ contribute to the increase of bending angle. But the contribution of second factor (mentioned by the reviewer) is much smaller due to the additional small parameter $H/Rc \ll 1$ and thus it was neglected.*

[Figure]

L747-749: "Although this effect is small, it can still cause a difference up to about 0.3% in the bending angles measured at the same impact height along the equator between the N-S and E-W directions". Where does the number 0.3% come from? How did you calculate it?

*Response: This estimate is based on two ROs located at Equator, one observing in E-W direction and the other in the N-S direction. This represents approximately the maximum possible difference in BA that can be achieved due to azimuth angle difference. We have added this explanation to the text.*

L748: Perhaps "at" instead of "along".

*Response: Suggestion accepted.*

L750: Better to say "as a function of impact parameter" instead of "as the function of the impact parameter".

*Response: Suggestion accepted.*

L760: "The azimuth angle effect can be corrected ...". I suppose you mean when comparing collocated BAs on impact heights. Please make that clear in the text.

*Response: We have rewritten the sentence: "In general, direct comparisons of BA from different RO missions are not physically meaningful unless the effect of azimuth angle is accounted for, typically through a model-based DD correction."*

L760-761: "... through double differencing (DD) using a model." I suggest to call it sampling error correction, not DD. Schreiner et al. (2019) used the term "sampling correction". The term DD in RO has been used for something else (to eliminate clock errors in the phase data by simultaneously observing two GPS satellites and using ground station data).

*Response: We agree that "double differencing" can mean more than one thing, We are following Tradowsky et. al. (2017) and Gilpin et al. (2019), who used the term to correct for biases between two datasets using a model, which is valid under the assumption that the biases in the model at one observation are the same as those collocated with the other observation. We have modified the sentence to read "The azimuth effect can be corrected through a type of sampling correction termed double differencing (DD) using a model."*

L784-785: "In RO data retrieval, it is commonly assumed that the occultation plane remains fixed throughout an occultation event and is anchored at the assigned occultation point, ...". My understanding is different: The centre of curvature and radius of curvature remains fixed, but the occultation plane is not assumed to be fixed. For each sample during an occultation a new occultation plane is calculated based on the fixed centre of curvature and the varying satellite positions. Thus, the occultation plane is assumed to change during the occultation, and the tangent point locations are different for each sample. We say that the tangent points are drifting. What is commonly assumed is that the surface of the Earth locally coincides with the surface of the sphere whose origin is at the centre of curvature. That is the assumption which does not hold exactly. It only holds in the direction of signal propagation at one instant during the occultation, the one that

defines the occultation point which is then fixed. It doesn't generally hold in any other direction or at another instant. Would you agree with this?

*Response: Thanks for pointing this out. We have revised the text as follows: "In RO data retrieval, a single reference sphere, defined by a fixed center and radius of curvature anchored at the occultation point, is typically used to approximate the Earth's surface throughout the entire RO profile. However, when the tangent point drifts horizontally, this reference sphere no longer accurately represents the local geometry of the Earth's ellipsoidal surface."*

L801: "can be corrected by adjusting the impact heights ...". Add "and altitudes".

*Response" We apply the correction only to impact heights. There may be other options for implementing the correction.*

L801-802: "... by a correction factor termed the sideways sliding correction." Use of the word "factor" implies that something should be multiplied by a factor, which is not the case. Text needs slight modification.

*Response: We have deleted "factor" from the sentence.*

L808: "Fig. 17: Difference in radius of curvature (dRc in km) along and across ray path ...". I suppose it is "across" minus "along" that is displayed in the figure, but the text is not clear on that. Please clarify what is subtracted from what.

*Response: Yes, the difference is "across" minus "along." The caption has been made clear.*

L812-815: "Tests of the impact of the sideways sliding correction by EUMETSAT and UCAR demonstrated that the vertical variation of the effect depends on how the nominal location or point of an occultation (termed occultation point by UCAR and georeferencing by EUMETSAT) is defined ...". Is there a reference that could be included here?

*Response: Jan Weiss in his presentation at the 2nd ROMEX Workshop at EUMETSAT 25-27 February 2025 showed a figure depicting the magnitude of the impact height correction vs. impact height for different definitions of occultation point. We have added a reference to this presentation and rewritten the sentence as "The magnitude of the correction varies with impact height depending on how the nominal location or point of an occultation (termed occultation point by UCAR and georeferencing by EUMETSAT) is defined (Weiss et al. 2025)."*

L827-831: "The sideways sliding correction results in a small reduction in the average positive C2-ERA5 BA and N biases in the UCAR-processed data by approximately 0.05% in the stratosphere. It also corrects negative biases associated with polar orbiting satellites, mostly in the tropics, by a similar amount. At higher latitudes, the effect on

observations from polar orbiters is negligible (less than 0.01%).". Please provide references.

*Response: This is based on tests we did but did not show the results here. However, we deleted this paragraph because the magnitude of the effect of the correction on BA and N is illustrated in Fig. 18 (now Fig. 19 in revised paper) and described in the following paragraph.*

L834-835: "... and the resulting C2-Spire bias is shown in Fig. 18. The reduction is smallest at 10 km ...". Lower than that, I suppose (but not shown). What is the typical altitude (or range of altitudes) of the tangent point for which the excess phase is 500 m?

*Response: Yes, there is no change (0 reduction) at the occultation point, which is defined where the excess phase is 500 m, corresponding to a typical tangent point altitude of ~4 km.*

L835: "... because of the definition of occultation point in the UCAR data". Insert "the" in front of "occultation point".

*Response: Suggestion accepted.*

L839: "Fig. 18: Bias of C2 relative to Spire for UCAR standard (solid profiles) ...". Perhaps skip the word "profiles" here.

*Response: Suggestion accepted.*

Figure 18: I suggest adding a plot showing the same for refractivity.

*Response: We added refractivity as suggested to this figure (now Fig. 19).*

L845: "For C2, the off-boresight angles exceeding 40° are mostly concentrated between 40-45°N/S". It would be interesting to see this in a figure.

*Response: Because the concentration of large off-boresight angles in the 40-45° N and 40-45° S latitude bands do not seem to play a role in larger biases there we have deleted this sentence.*

L852-857: "An average of 35,000 RO profiles per day from 13 different RO missions from the United States, Europe, and China are being used in NWP models at major international centers to study how different numbers of RO profiles affect the analyses and forecasts. This paper evaluates the characteristics of the ROMEX data used in these experiments, with emphasis on the three largest datasets, COSMIC-2, Spire, and Yunyao." My understanding is that the data analyzed in this paper are not the same as the data provided to the NWP centers for all missions (C2 may be an exception). There may be differences in the biases and uncertainties due to different processing, and I have pointed to some examples in my comments above (L138-141). It is fine that this paper

analyses the UCAR-processed data, but I don't think you can write that it is generally the ROMEX data that are being used in experiments at NWP centres. Please modify text.

*Response: Correct, most NWP experiments have used the EUMETSAT-processed ROMEX data. We replaced "used in these experiments" by "processed by UCAR."*

L884-885: "This apparent bias is investigated and found to be a result of their different orbits.". I think the statement is correct that it is due to different orbits (or orbit configurations). However, I don't see any real evidence in the paper. I see a reduction in C2 bending angle biases when doing DD (sampling error correction) in Fig. 16, but that was already shown by Schreiner et al. (2019). Could more evidence that the reason is to be found in the different orbit configurations/azimuths be provided (cf. my comments to L724-725 and L747-749)?

*Response: We have added a figure showing the different azimuth angles (new Fig. 15) and the opening statement of Section 5.2 is "The small positive biases of C2 relative to Spire and other ROMEX missions in BA between 10 and 35 km result from their different orbits around the non-spherical Earth."*

L889: "... is different azimuth or viewing angles on the average ...".  This is the first time "viewing angles" are mentioned, and they are not discussed further. Please clarify.

*Response: We deleted the reference to viewing angles.*

L895: "... sideways sliding of the occultation planes during occultations.". I suggest to say "... sideways sliding of tangent points" (cf. comment to L784-785).

*Response: We have referred to this effect as the "sideways sliding of the occultation plane and tangent point" in the revised document. Josep Aparicio used both terms in his November 15, 2024 presentation.*

L896: "This sliding results in different radii of curvature of Earth ...". The sliding does not result in different radii of curvature, they are already there. Please rephrase.

*Response: This text has been revised to: "The second source is the horizontal sliding of the RO tangent point, which leads to a height difference between its position relative to the Earth's ellipsoid surface and the reference sphere. This difference results in a positive bias of ..."*

L895-896: "... and different impact parameters ...". The impact parameters are not changed, only the height of tangent points (which are more accurately referenced to the surface of the ellipsoid instead of to the surface of a reference sphere). Please rephrase.

*Response: Rephased as stated in the previous response.*

L897-899: "... and creates a positive bias of about 0.05% in the UCAR-processed C2 bending angle (BA) and refractivity (N) observations in the stratosphere compared to those of the polar orbiters". The positive bias seems less than 0.05% in Fig. 18. (perhaps half of that at 30 km where the difference between C2 and Spire is largest).

*Response: We changed this to "and creates a positive bias of up to 0.05%" in the revised paper.*

L899: "The sideways sliding effect was identified and discussed by Josep Aparicio in November 2024. It can be easily corrected in the processing of the RO data by applying a correction to the impact height." I don't agree with this statement. My understanding is this: The impact height is not involved in the processing of RO data (the impact parameter is involved, but it is not changed). The impact height is used when we compare bending angles on a vertical scale (where we then subtract the radius of curvature from the impact parameter), and it is here we need to make a correction. Would you agree with this?

*Response: We may not fully understand the exact point of disagreement, but we would like to clarify our perspective. In the context of RO data processing, the impact height serves as the argument of the BA function. It is not an arbitrary choice but a direct result of the processing itself. Therefore, whether or not the impact height is considered as being "involved" in the processing may be more of a semantic distinction. We choose to use impact height as the argument for the BA because it facilitates direct comparisons of BA produced by different data processing centers and allows for meaningful statistical averaging. Such comparisons would not be straightforward if BA were instead expressed as a function of impact parameter.*

*In his presentation on 15 November 2024 (personal communication), Josep Aparicio applies his sideways sliding correction dR to the impact height h, where $h = a - R_c$ is the uncorrected impact height, a is the impact parameter, $R_c$ is the local radius of curvature, and h' is the corrected impact height*

$h' = h - dR = a - R_c - dR$ . $\qquad\qquad\qquad\qquad$ (1)

*Eq. (1) may be written in terms of a corrected impact parameter*

$h' = a' - R_c$ $\qquad\qquad\qquad\qquad\qquad\qquad\qquad$ (2)

*We apply the correction in the UCAR CDAAC revised processing by correcting the impact height according to (1). However, we also supply the correction dR with the processed data in case users wish to apply the correction in a different way.*

*In contrast, assimilation systems typically operate using the impact parameter, a. However, this curvature radius (or reference sphere) is a simplified representation of the Earth's surface that applies uniformly for the entire occultation profile. As discussed in the paper, this introduces inaccuracies in the estimated ray heights over the true Earth's surface. Therefore, correcting for impact height effectively translates to a correction of the impact parameter for the purpose of assimilation.*

*If, as suggested by the referee, the impact parameter is left unchanged, this would indeed preserve its formal geometric meaning as the ray's miss-distance to the center of sphericity. However, the error (bias) related to the miss-representation of the reference ellipsoid by using the single center of sphericity will propagate into the data assimilation.*

*We acknowledge that formally correcting the impact parameter is not the only way to account for Earth's oblateness in assimilation systems; alternative methods are indeed under discussion. However, such assimilation-specific adjustments are beyond the scope of this paper.*

L906-907: "Code and data availability. The ROMEX data are available free of charge through ROM SAF under the ROMEX terms and conditions.". Please give precise information which ROMEX data were used. Are you sure the data processed by UCAR and STAR are on the ROM SAF server? I was not able to find it. There are different datasets in different versions. Which ones were used for this study?

*Response: As stated in L137-139 in the original paper, we have analyzed the UCAR-processed ROMEX data, which are similar to the EUMETSAT data in most respects. The UCAR-processed ROMEX data are not yet on the ROM SAF server; but they should be there very soon. They are available from UCAR upon request from interested users. We have added this information to the "Code and data availability" section.*

L918: "JS assisted RA and JS by ...". There are two JS's.

*Response: Full last names are now spelled out to avoid any confusion.*

L980: I think "Chen" should be "Cheng".

*Response: Yes, corrected.*

L985: "Presentation at the 1st ROMEX Workshop April 17, 2024 at EUMETSAT, ..." I think you mean: "Presentation at the 2nd ROMEX Workshop February 27, 2025 at EUMETSAT, ..."

*Response: Yes, corrected.*

L1077-1081: The reference can be updated since the final revised paper has been published.

*Response: Yes, updated.*

Supplement:

There are detailed discussions in the figure captions of Figs. S1.2-S1.8, which corresponds to Fig. 12 in the paper. Wouldn't it be better to have these discussions in the text in the paper?

*Response: We added a reference to the detailed discussions in the Fig.12 caption: "Larger versions of the panels with some comments on each level are presented in the Supplement (S1)." There are quite a few comments and to add them all in the caption of Fig. 12 would make a very long caption.*

More information on what is shown in the three panels of Fig. S2.1 should be added to the figure caption.

*Response: We have added comments to the caption of Fig. S2.1.*

I think some of the N bias maps (Figs. S2.7 and S2.8) depend significantly on the use of a climatology in the statistical optimization at high altitudes (cf. my comment to L138-141.).

*Response: We agree, but we are evaluating the UCAR-processed data and hence use the statistical optimization used in the UCAR processing. The effect of the statistical optimization on the refractivities is mainly above 40 km, and we do not discuss refractivities above this level in this paper.*

Fig. S3.1: Why show this figure again?

*Response: We want to have the vertical profiles of the CSY uncertainties in the same place as the global maps of the CSY uncertainties so the reader has all of them in one place.*

Fig. S6.1: It seems that N is affected by climatology above 30 km, and there is no positive bias relative to ERA5 at 40 km, as there is in bending angle (Fig. S5.1). Could you comment on it in the Supplement (or in the paper)?

*Response: The relationship between BA and N is non-local so there is not a one-to-one correspondence between the BA biases and the N biases at all levels. We added a comment to the caption of Fig. S6.1.*

Fig. S6.6: "All three missions have similar distributions.". I think that is an overstatement. I see significantly larger positive biases for C2, which is also seen in Fig. S6.1. Do you have an explanation for that? (cf. also my comment to Figure 18 in the paper).

*Response: We have deleted the statement about similar distributions. We discuss the complicated C2 biases vs. ERA5 biases in the paper.*

*End of responses to Reviewer 2*

---

## Referee Report (RR1)

**Review Summary:**

The authors have diligently addressed the comments of this reviewer. There are still some clarifications requested.

**Detailed Comments:**

Line 250 (track changes version): the word "Perceived" here does not seem quite right. A better word could be "Resulting", or even no word at all.

Line 331 (track changes version): "These are ..." seems to directly repeat the previous phrase?

Figure 7 (original line 502): While the authors are correct that temperature uncertainty and water vapor uncertainty of ERA5 are expected to differ, there is no attempt by the authors to compare their determination of bending angle uncertainty with previous determinations of ERA5 uncertainty. Discrepancies between these results and the ERA5 literature may be present and require some explanation. Comparisons between ERA5 and radiosonde below 10 km altitude show roughly constant or gradually decreasing relative humidity uncertainty of ERA5 towards the surface, assuming radiosonde as "truth" (e.g. Gamage et al., Figure 2; Virman et al., Figure 4). Bending angle is sensitive to specific humidity. A constant relative humidity uncertainty with altitude would result in an increasing specific humidity or bending angle uncertainty with decreasing altitude as seen in Figure 7, but the sharp uncertainty reduction in the lowest ~1-2 km is not consistent with the radiosonde comparisons. What are possible explanations for the discrepancy between these results and the prior literature? Is there a sharp reduction in ERA5 temperature uncertainty near the surface that counteracts the increasing specific humidity uncertainty towards the surface in ERA5, so that bending angle uncertainty shows the pattern in Figure 7?

**References:**

Gamage et al. (2020), A 1D Var Retrieval of Relative Humidity Using the ERA5 Dataset for the Assimilation of Raman Lidar Measurements, Journal Of Atmospheric And Oceanic Technology, doi: 10.1175/JTECH-D-19-0170.1.

Virman et al. (2021), Radiosonde comparison of ERA5 and ERA-Interim reanalysis datasets over tropical oceans, Tellus A, doi: 10.1080/16000870.2021.1929752.

---

## Author Response (AR3)

**Response to second two reviews**

We thank the two reviewers for their careful reviews of our revised paper and their helpful suggestions. Our responses are included below in italics.

**Report #1**

Review Summary:

The authors have diligently addressed the comments of this reviewer. There are still some clarifications requested.

**Detailed Comments:**

Line 250 (track changes version): the word "Perceived" here does not seem quite right. A better word could be "Resulting", or even no word at all.

Response: "Perceived" is removed as suggested in a revised sentence.

Line 331 (track changes version): "These are ..." seems to directly repeat the previous phrase?

Response: This was a typo. It has been corrected.

Figure 7 (original line 502): While the authors are correct that temperature uncertainty and water vapor uncertainty of ERA5 are expected to differ, there is no attempt by the authors to compare their determination of bending angle uncertainty with previous determinations of ERA5 uncertainty. Discrepancies between these results and the ERA5 literature may be present and require some explanation. Comparisons between ERA5 and radiosonde below 10 km altitude show roughly constant or gradually decreasing relative humidity uncertainty of ERA5 towards the surface, assuming radiosonde as "truth" (e.g. Gamage et al., Figure 2; Virman et al., Figure 4). Bending angle is sensitive to specific humidity. A constant relative humidity uncertainty with altitude would result in an increasing specific humidity or bending angle uncertainty with decreasing altitude as seen in Figure 7, but the sharp uncertainty reduction in the lowest ~1-2 km is not consistent with the radiosonde comparisons. What are possible explanations for the discrepancy between these results and the prior literature? Is there a sharp reduction in ERA5 temperature uncertainty near the surface that counteracts the increasing specific humidity uncertainty towards the surface in ERA5, so that bending angle uncertainty shows the pattern in Figure 7?

**References:**

Gamage et al. (2020), A 1D Var Retrieval of Relative Humidity Using the ERA5 Dataset for

the Assimilation of Raman Lidar Measurements, Journal Of Atmospheric And Oceanic Technology, doi: 10.1175/JTECH-D-19-0170.1.

Virman et al. (2021), Radiosonde comparison of ERA5 and ERA-Interim reanalysis datasets over tropical oceans, Tellus A, doi: 10.1080/16000870.2021.1929752.

Response: This figure is now Fig. 11. This is a good question. We do not think there is any discrepancy between the uncertainty profiles of the ERA5 bending angle (BA) in our study and uncertainty estimates of temperature and water vapor. ERA5 BA uncertainties are different from the ERA5 uncertainties of temperature or water vapor as discussed in the Gamage and Virman references. The maximum in BA uncertainty at about 3 km impact height and the sharp reduction below (Fig. 11 of our revised paper) is a common feature of uncertainty estimates of RO BA (e.g. Anthes and Rieckh (2018), Semane et al. (2022), Ho et al. 2023-references in the paper). We do not know of any references that explicitly explain the maximum of BA uncertainties around the top of the atmospheric boundary layer (ABL), but it is clear that it is related to the maximum vertical gradient of refractivity at the ABL top, as discussed by Ao et al. (2012) and Xie et al. (2012) and shown in Fig. 2 of the Xie et al. paper (reproduced below). The BA is related to the vertical gradient of refractivity, which can be extremely large at the top of the ABL.

This maximum in BA uncertainties at the top of the ABL is present in RO observations and models, which compute the BA from vertical gradients of refractivity using a forward model. Mean differences in the model ABL height and the ABL height estimated by the RO observations as found by Ao et al. (2012) may also contribute to the uncertainties.

The maximum in uncertainties in the observed and model RO profiles is not present in the temperatures and water vapor, as noted by the reviewer.

We have added an explanation of the large uncertainties and biases in models in Section 1.1 (lines 217-222) of the revised paper.

Fig. 2. Typical ABL structure in one VOCALS radiosonde (black) with the near-coincident COSMIC-RO (blue) and ECMWF analysis (green) profiles. (a) Radiosonde specific humidity and temperature; (b) ECMWF analysis specific humidity and temperature; The boxes in (a, b) indicate the cloud region with relative humidity exceeding 94%; (c) refractivity of radiosonde (black), COSMIC RO (blue) and ECMWF analysis (green) as well as simulated refractivity retrieval (black-dotted) based on the simulated radiosonde bending; the ABL heights are listed in the upper right corner, respectively; (d) simulated bending angle of radiosonde (black), ECMWF analysis (green) as well as the standard (blue-solid) and high-resolution (cyan) COSMIC RO bending angles.

Reference (the other references in Response presented in the paper): Xie, F. et al. 2012: Advances and limitations of atmospheric boundary layer observations with GPS occultation over southeast Pacific Ocean. Atmos. Chem. Phys., 12, 903–918, www.atmos-chem-phys.net/12/903/2012/doi:10.5194/acp-12-903-2012

**Report #2**

I am happy with most answers to my questions and changes to the manuscript. However, there are still a number of minor issues which I think should be addressed before I can recommend the paper for publication.

Below, I have indicated each issue by the original line numbers from my first review, which were repeated in the authors response, and with the corresponding line numbers from the revised manuscript in parenthesis.

L33-34(L33-34): The abstract should relate to the data that are analysed and presented in the main paper. Thus, I suggest to remove or qualify the sentence in the abstract that "They are similar on the average in the region of overlap (45°S-45°N)". The sentence is not correct for the data that are presented in Fig. 8 (between 10 and 15 km). Although I understand that a later

Yunyao processing has remedied the differences, the data that are presented are the ROMEX data that will be made available to the community (as I understand it).

Response: We prefer to leave this sentence in the Abstract. This statement is an important conclusion (it means that all RO missions can be treated similarly in NWP models) and is supported by the comparison of all 14 RO missions in Fig. 9 (revised version of paper), and the three CSY missions in Fig. 12 (revised paper) over most vertical levels between 0-60 km. The possible exception in the blip in the Yunyao uncertainties between 10-15 km and the reason for this blip is explained in the paper.

L87-88(L88-89): Both of the added references are not peer-reviewed, and one is a personal communication. It should be possible to find better references (the text says "widely studied").

Response: Schreiner et al. (2020) is peer reviewed. We inadvertently left the Bowler (2020) reference out of the revised paper; it is now included.

L91(L92): In the list of references (lines 1056-1063 in the revised manuscript), Cheng (2024) refers to the 1st ROMEX workshop, and Cheng (2025) refers to the 2nd ROMEX workshop. Thus, I think it should be (Cheng, 2024) here.

Response: Thank you; we changed the year to 2024.

L106-107(L108-113): There should be better (peer-reviewed) references than the ROPP guide, e.g., papers that it is based on.

Response: The ROPP guide referenced here gives a thorough, detailed discussion of the RO processing, much more detail than any peer-reviewed paper, and is readily available. It is the original and carefully reviewed document (internally) and we prefer to retain it in our paper.

L138-141(144-151): Refractivity biases above 40 km are shown and discussed in the Supplement. It would be relevant there to mention that there can be differences in refractivity biases at high altitudes due to different approaches in statistical optimization.

Response: We have added the following sentence to L718 in the revised paper "We note that the N biases above 40 km are affected by the statistical optimization, which can vary with different processing centers."

L144-146(L153-L158): It seems not right to measure/define "depth" in percent here. In line 156 of the revised manuscript it says "The penetration depth (lowest level reached)...", which makes more sense to me. I think the paragraph would be consistent by removing the words "depth" in lines 153 and 154, and the word "rates" in line 157.

Response: We have changed the wording in Lines 156-159 to read: "We estimate the lower tropospheric penetration depths (lowest level reached) of the RO profiles, the standard deviation of random errors (uncertainties), and biases. The penetration depths depend on the cutoff

criteria used in the processing, and so their comparison among different missions should be done with the same processing center."

L180-183(L184-188): The title of the added reference to (Schreiner et al., 2020) is incorrect (line 1204 in the revised manuscript). It should be "COSMIC-2 Radio Occultation Constellation: First Results". The added reference to (Ho et al., 2023) is not in the reference list.

Response: Thank you. We corrected the Schreiner et al. 2020 reference and added the Ho et al. (2023) reference

L184-189(L189-194): I do not see the link to irowg.org/romex-events-meetings/ in the revised manuscript (as written in the response to my first review), but it is anyways not straightforward to find the presentations at the link. The reference to (Shao et al., 2025) is to the IROWG-10 meeting (which was before the second ROMEX workshop but does contain some information on what happened at the first ROMEX workshop). In addition to the reference to Shao et al., I suggest to include the links to both ROMEX workshops here.

Response: We added the link to the ROMEX workshops as suggested.

Figure 1: As I understand it, separate captions for each panel is not allowed according to the journal policy. I think the problem is best solved by renumbering 1a to 1e as figures 1 to 5. I don't see much relation that justifies keeping them as one figure.

Response: We have renumbered the different panels of Fig. 1 as suggested.

Figure 1a: The underlying world map with continents should be explained or removed. This is a plot in latitude vs local time where a world map makes very little sense.

Response: The faint background of the continents is present to give the reader an idea of the scale of the orbital distributions (for example, the scale of the gaps in local time coverage of Yunyao). We have added a statement to this effect in the figure caption.

L374(L391): I don't understand why is it called 3CH QC. There is no reference to 3CH QC in section 2.3. Perhaps it could be called "final QC" as it is referred to in section 2.3.

Response: We changed "3CH QC" to "final QC" as suggested.

L468(L492): The sentence has not been changed as written in the response to my first review.

Response: This was an oversight. The revised sentence in L511 (with the new Figure number reference) is: "The biases of all ROMEX missions with respect to ECMWF analyses appear very close to zero on this scale of the x-axis (Fig. 9), but a closer look shows a small negative bias of approximately -0.1% in most ROMEX missions between 10 and 35 km as shown in Fig. 10a."

Figure 8: Perhaps say 'barely visible' in the caption instead of 'not visible'.

*Response:* Suggestion accepted in the caption to Fig. 12 (the new Figure 8).

L651-652(L696-697): I am very sceptical that ERA5 (including forward modeling errors) could have biases up to -15% at 3 km impact height. In Section 1.1 it says "such as errors in the coefficients of the refractivity equation". I suppose that cannot create what we see in Fig. 12a (I think letters should be added in Fig. 12 panels, as they are in Fig. 9 panels). I don't know if the use of a 1D model could be the reason. If so, it would be good to mention this explicitly here.

Response: Fig. 12 is now Fig. 16 and letters have been added.

These are mean differences between the CSY BA observations and BA computed from a forward model from the ERA5 data. We do not claim that the large (15%) biases seen over the tropical West Pacific are biases in the ERA5 data, they could be mostly due to the biases in the RO data. Also, it is important to note that the biases and uncertainties in the model BA do not necessarily imply biases and uncertainties of similar magnitudes in the model temperature or water vapor. The BA are a function of the vertical gradient of these model variables, and may also arise from systematic errors in the forward model, such as errors in the coefficients of the refractivity equation. We have added this explanation to lines 217-222 in the revised paper. However, the reasons for the large biases in the tropical West Pacific, which are present in all three CSY missions (Fig. S33 of Supplement), are uncertain and require further investigation.

L672-674(L717-719): The sentence could be misunderstood as "... mean and standard deviation of differences between BA and N". I think the whole sentence could be removed without loss of information.

Response: We removed the sentence as suggested.

L694(L749): My suggestion to say "biased relative to" was accepted in the response to my first review, but the sentence has not been changed.

Response: Another oversight—the sentence has now been changed as suggested.

L724-725(L778-782): I'm happy with the inclusion of the new Fig. 15, which shows a rather broad distribution of azimuth angles for both C2 and Spire. Thus, the text saying "predominantly oriented in an east-west (E-W) direction" seems a bit imprecise to me. I would rather say "predominantly oriented within  $\pm 45^{\circ}$  of the east-west (E-W) direction ... occultation planes generally oriented within  $\pm 45^{\circ}$  of the north-south (N-S) direction".

Response: We have made this change as suggested (L810-813) of revised paper).

L746(L807-808): Although I cannot immediately see the implication for the bending angle in the response, I think the authors are right. I suggest to say "... due to traversing a slightly longer path within an atmospheric shell ...".

Response: We have made this change as suggested (L843 of revised paper).

L801(L864): I don't understand how the correction was only applied to impact heights (as noted in the response to my first review). In the new Fig. 19 both BA and N corrections are shown. It is not clear how the N correction was applied as a function of altitude in that figure.

Response: The correction is applied to the impact height and then transferred to the impact parameter (by retaining the local radius of the best-fitted sphere to Earth's surface at the occultation point). The Abel inversion is subsequently performed to derive the refractivity, followed by the standard procedure to obtain the MSL altitude. Thus, the correction is applied only to the impact height, but its influence naturally propagates to the altitude.

We have added the following sentence to L904: "Assigning the retrieved BA to an adjusted impact height is effectively equivalent to modifying the BA for a given impact height. Consequently, this adjustment further influences the refractivity as a function of altitude through the subsequent Abel inversion."

L808(L871): "Difference ... minus". The sentence should be rephrased.

Response: This is the Rc across the ray path minus the Rc along the ray path so we think the wording is OK. We added parentheses (across minus along) which may help.

L827-831(L890): Typo: B should be N.

Response: Typo corrected.

L834-835(L891-893): I suggest to say "The reduction is smallest in the lower troposphere ...".

Response: We said the reduction is smallest at 10 km because that is the lowest level shown in Fig. 23 (old Fig. 19). However, the suggested change is better, and we have accepted it.

Figure 18 (Figure 19): It is unclear how the correction to N was applied (see earlier comment).

Response: Please see response to L801(L864) above.

L899(L960-962): As I understand it, the sideways sliding correction was applied to both bending angle as a function of impact height and refractivity as a function of altitude (Fig. 19). I'm unsure if this last statement refers to what was actually done in the paper, or if it is something that has been done only in a revised processing mentioned in the response (which is not otherwise mentioned in the paper). I suggest to mention only what was done in the paper.

Response: The last statement is correct (L1002) in revised paper), and affects both the BA as a function of impact height and N as a function of altitude as described above. It is what is done in this paper.

L906-907(L967-968): The data should be put on the ROM SAF ROMEX server. It was not there the last time I checked.

Response: We agree and are disappointed as well. We sent the UCAR-processed data to EUMETSAT in April 2024 and have repeatedly asked them to put the data on the ROM SAF ROMEX server, including numerous times since we submitted our paper in August 2025. Despite many assurances that the data would be there "soon," it has not been done and we have given up. We have changed the wording to: "The ROMEX data processed by EUMETSAT are available free of charge through ROM SAF under the ROMEX terms and conditions. Further information is available at <a href="https://irowg.org/ro-modeling-experiment-romex/">https://irowg.org/ro-modeling-experiment-romex/</a>. The ROMEX data processed by UCAR are available from UCAR under the ROMEX terms and conditions."

**Supplement:**

Fig. S2.1: I suggest to remove new sentence saying: "Precise version of top label is <<(CSY-<ERA5>)>>/<<ERA5>>. Same for Figs. S2.2-S2.8". I don't think it is explained anywhere what it means.

Response: We prefer to keep this brief note. It is defined in lines 44-45 under Fig. S1 in the Supplement.